# MOTION-AWARE SURFACE SMOOTHING FOR AVATAR REPRESENTATIONS FROM MONOCULAR VIDEOS

## ABSTRACT

3D Gaussian Splatting (3DGS) has become a popular representation for 3D avatar modeling due to its fast training and real-time rendering. However, state-of-the-art methods struggle to generalize from monocular videos and often fail to recover realistic geometry. We introduce a motion-aware surface smoothing framework to improve 3DGS for learning from monocular human videos. Our method regularizes the training of Gaussian parameters, modulates the Adaptive Density Control (ADC) for improving surface quality, and supervises Gaussian motions under unseen camera viewpoints. This enforcement of surface smoothness yields superior geometry contours and higher-fidelity rendering. Across five public datasets, including MVHumanNet, DNA-Rendering, ActorsHQ and outdoor videos, our approach consistently outperforms prior methods in novel view synthesis, novel pose animation, and 3D shape reconstruction. Code will be published upon acceptance.

## 1 INTRODUCTION

Reconstructing an avatar representation from videos plays a crucial role in many applications (Wu et al., 2019; Bagautdinov et al., 2021; Lombardi et al., 2021; Hu et al., 2022; Song et al., 2023). Traditional methods achieve this by using parametrized human body templates (Loper et al., 2015) or rigging a surface mesh to an underlying skeleton (Bagautdinov et al., 2021; Liu et al., 2021). However, they often rely on dense, synchronized multi-view inputs, which is not practical for many outdoor scenarios. Also they are difficult to optimize due to their explicit usages of vertices and faces and are not adaptive to dynamic cloth details like wrinkles. Recently, neural radiance fields (NeRF) (Mildenhall et al., 2020) have enabled realistic avatar modeling from multi-view video sequences. They either transform the human bodies of each frame to the canonical space with T-pose (Wang et al., 2022; Jiang et al., 2022; Li et al., 2022), or condition the NeRF models on relative positions of each human body part (Su et al., 2021; Noguchi et al., 2021; Su et al., 2022; Song et al., 2024). The more recent 3D Gaussian Splatting (Kerbl et al., 2023a) further empowers the neural-based 3D human rendering to achieve fast training and real-time image synthesis with high image quality (Qian et al., 2024; Moreau et al., 2024; Kocabas et al., 2024a). The core idea of 3DGS-based human modeling is to represent a 3D human body with a set of learnable Gaussian parameters in canonical space, which are then transformed to the per-frame observation space with the pre-computed SMPL skeletons for novel view rendering.

Despite their success in multi-view image synthesis, as depicted in Fig. 1 (a), the existing 3DGS-based methods cannot generalize well when learning from monocular videos. We hypothesize that the limited generalization capabilities of these approaches come from their noisy and low-fidelity geometry approximation. Conceptually, they are trained to overfit the only given camera view instead of learning a consistent geometric human representation. As a result, they can faithfully reproduce the training image sets but are prone to distorting the detailed structures (e.g. wrinkles) and producing artifacts near the shape boundaries when testing under different camera views.

In this paper, we assume an ideal 3D surface to be both smooth and thin, based on which we explore a novel Motion-Aware Surface Smoothing for Avatar Representations (MASSAR) to address the aforementioned limitations. Similar to former methods, we define a set of learnable Gaussians in the canonical space. Taken a monocular video as input, we first render a depth map for each frame and correspond each pixel of the per-frame depth map with the closest Gaussian along each pixel's ray direction. To improve smoothness of estimated surfaces, we encourage the 3D Gaussians to distribute

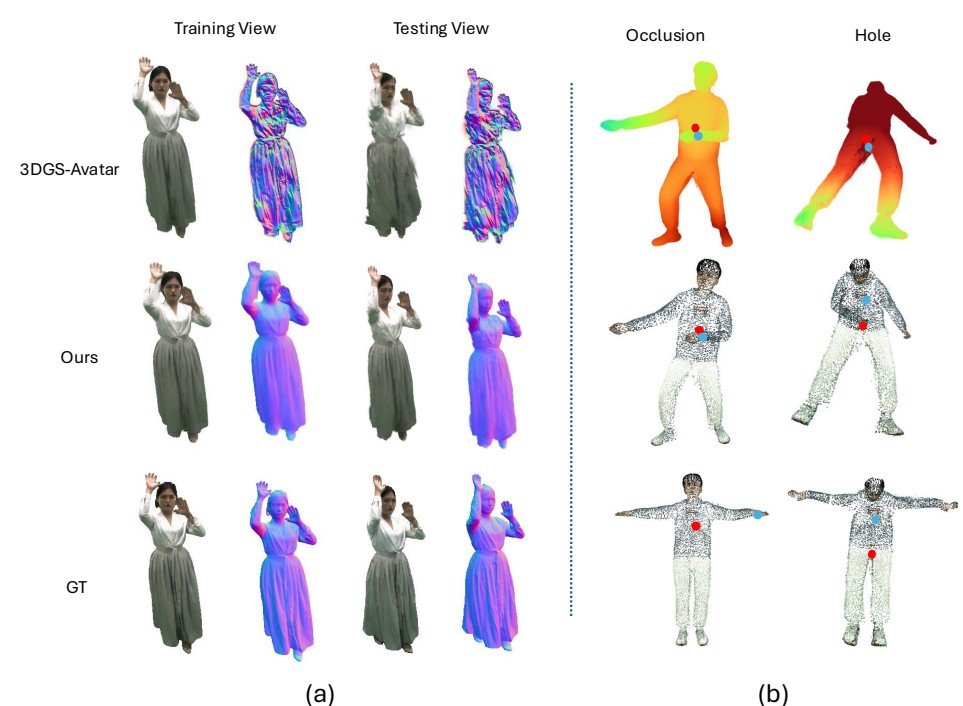

Figure 1: **Teaser.** (a) Existing methods struggle with monocular videos, often yielding unrealistic geometry. Our depth handling reduces such artifacts. (b) Due to human dynamics, neighboring pixels (blue and red) may correspond to different geometric elements—which we model explicitly.

smoothly in space by smoothing the rendered depth maps. However, as exemplified in Fig. 1 (b), the nearby pixels in some frames probably match the Gaussians which are far from each other in the canonical space due to the self-occlusions and the holes occuring during motions. Thus, we measure a smoothness value for each depth pixel with the distances between canonical Gaussians to cater for the dynamics nature of articulated humans. The benefits of the smoothness computations are threefold: 1) Regularizing the training of Gaussians; 2) Modulating the Adaptive Density Control for stable surfaces; 3) Supervising Gaussian motions under unseen camera views.

Besides the motion-aware surface smoothing module, we effectively integrate the rigid human motion with its non-rigid counterpart to approximate multi-scale deformation effects. Specifically, we first apply a learnable multi-layer perceptron (MLP) for the non-rigid motion to generate per-Gaussian offsets in canonical space, followed by performing the inverse linear blending skinning (LBS) for large-scale rigid motions. Our motion-aware surface smoothing achieves state-of-the-art results on five widely used datasets across the tasks of novel view synthesis, unseen pose animation and 3D surface reconstruction. Our key contributions are:

- We introduce the motion-aware surface smoothing for avatar representation (MASSAR) learning by explicitly measuring the smoothness of depth maps.
- We develop three strategies to utilize the surface smoothing for improved geometry modeling and superior generalizations to monocular videos within the 3DGS framework.
- We improve state-of-the-art methods in novel view synthesis, unseen pose animation, and shape reconstruction simultaneously.

## 2 RELATED WORK

**Articulated Human Modeling.** Since the emergence of neural radiance field (NeRF) (Mildenhall et al., 2020), many efforts explore to use neural representations to produce high-fidelity digital humans. Given a set of multi-view synchronized videos, some early NeRF-based methods (Su et al., 2021; Noguchi et al., 2021; Su et al., 2022; 2023; Song et al., 2024) transform input query points into

the relative coordinates of each part and estimate the density and color components locally. Later works (Peng et al., 2021a; Wang et al., 2022; Li et al., 2022; 2023c; Guo et al., 2023; Song et al., 2025b) improve modeling generalizations by formulating a static radiance field with a canonical T-pose and then warping the human body from each frame to the canonical space. To this end, other methods (Weng et al., 2022; Yu et al., 2023; Song et al., 2025a) further model the skeletal rigid and matched non-rigid motion field simultaneously to advance detail synthesis in monocular videos. (Guo et al., 2023; 2024; Shin et al., 2024; Tan et al., 2025; Song et al., 2025a) additionally incorporate the Signed Distance Function (SDF) representation to recreate realistic human shapes.

Recently, 3D Gaussian Splatting (3DGS) enabled fast training and real-time inference with realistic image synthesis for human avatars (Qian et al., 2024; Shao et al., 2024; Svitov et al., 2024; Kocabas et al., 2024a; Moon et al., 2024; Moreau et al., 2024; Hu et al., 2024b; Kocabas et al., 2024b; Lei et al., 2024; Zhan et al., 2025). A core technical component of these methods is to define a set of learnable Gaussians in the canonical space and use the Linear Blend Skinning (LBS) and SMPL skeleton to realize the rigid deformations. Despite these progresses, none of them take the geometry modeling into account. Different from GoMAvatar (Wen et al., 2024), which aims to reproduce precise surfaces with the Gaussians-on-Mesh (GoM) representation, we enhance the geometry representations by enforcing a simple and effective geometric smoothness constraint with much faster training speed. Compared to all these algorithms, we consistently provide superior results in image synthesis and geometric clues across several public datasets.

**General Surface Reconstruction.** The success of NeRF attracts significant interest in surface reconstruction with neural rendering technology (Wang et al., 2021; Yariv et al., 2021; 2023), which learn to map the input coordinates to corresponding occupancy fields or SDF values with a multilayer perceptron (MLP). Subsequence works (Fu et al., 2022; Yu et al., 2022) utilize geometry priors to further refine reconstruction accuracy, while others (Li et al., 2023b; Wang et al., 2023a;b; Reiser et al., 2024) employ hash encoding (Müller et al., 2022) to have a more expressive scene representation and improve rendering speed. More recently, some approaches aim to integrate 3DGS for surface learning. To improve geometry synthesis, Sugar (Guédon & Lepetit, 2024) and 2DGS (Huang et al., 2024) approximate 3D Gaussians with 2D Gaussians while (Dai et al., 2024; Wu et al., 2024b) optimizes Gaussian learning with a pre-trained normal prior. Later on, GoF (Yu et al., 2024b) constructs a Gaussian opacity field upon 3D Gaussians and identifies level sets to determine surfaces. Additionally, some works (Yu et al., 2024a; Lyu et al., 2024; Chen et al., 2023a; Zhang et al., 2024b) jointly optimize the trainable Gaussian parameters and neural SDF while PSGR (Chen et al., 2024a) further includes the prior of multi-view geometric consistency to achieve more precise surface extraction. Under the context of animatable avatars, GoMAvatar combines Gaussian splats with deformable meshes to boost geometric outputs. Fundamentally, different from these works, our method enforces the local surface smoothing by both a regularized loss term and a geometry-aware Adaptive Density Control (ADC). This strategy yields a continuous and compact human surface and, in turn, advances the rendering generalization without introducing significant costs. Sec. A for additional discussions.

## 3 METHOD

We aim to create a 3D avatar from a monocular video. To achieve this goal, we propose a novel motion-aware surface smoothing and integrate it into the 3D Gaussian Splatting (3DGS) framework. Our key technical contribution is to regularize the training of learnable Gaussian parameters and modulate the Adaptive Density Control of 3DGS by a smoothness measure on the depth map rendering.

### 3.1 PRELIMINARIES

**3D Gaussian Splatting.** The 3DGS framework approximates a 3D scene with a set of Gaussian primitives and achieves state-of-the-art rendering quality with real-time speed. Each 3D Gaussian is represented by its mean $\mu \in R^3$, covariance $\Sigma \in R^{3 \times 3}$, opacity $\alpha$ and view-dependent color c as

$$G(x) = \alpha \cdot e^{-\frac{1}{2}(x-\mu)^T \Sigma^{-1} (x-\mu)},  \tag{1}$$

where $x$ is a 3D position. In practice, we employ a 3D vector $\mathbf{s} \in R^3$ as the scaling coefficients and a 4D vector $\mathbf{q} \in R^4$ as a quaternion to reconstruct the covariance matrix $\Sigma$. During the rendering process, the 3D Gaussians are projected onto a 2D image plane and then the pixel color $C$ is

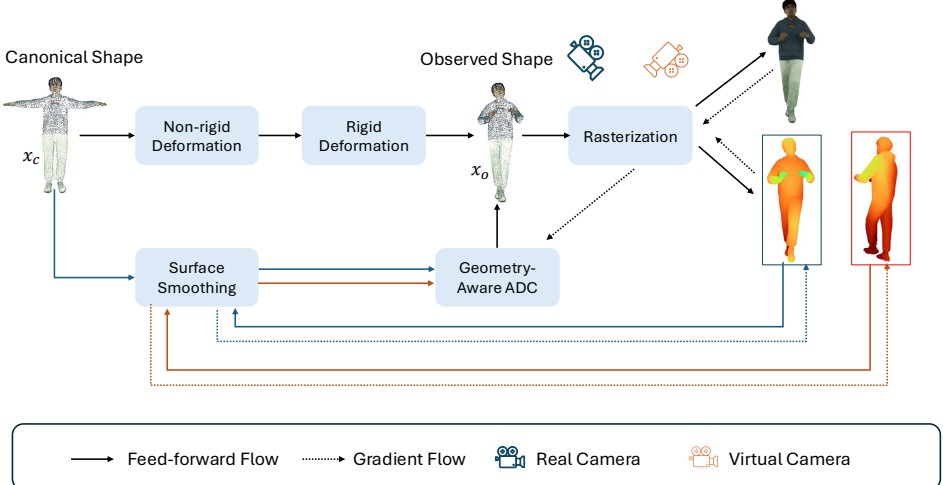

Figure 2: **Framework Overview.** The canonical 3D Gaussian at $x_c$ is mapped to the observation space $x_o$ through non-rigid and rigid deformations, then rendered via differentiable rasterization. Our surface smoothing module regularizes parameter training by leveraging rendered depth maps and canonical Gaussians, and guides their placement and density for more accurate geometry. Virtual camera settings further provide supervision for unseen motions.

computed by sorting and $\alpha$-blending 3D Gaussians. The 3D Gaussian parameters $\{\mathcal{G}\}$ are optimized with reconstruction loss functions. Based on the gradients of each Gaussian, the 3DGS framework also utilizes the Adaptive Density Control (ADC) to accommodate the underlying geometry and appearance patterns and then improve the scene representation accuracy.

Gaussian Opacity Field (GoF) (Yu et al., 2024b) further improves 3DGS by explicitly evaluating the ray-Gaussian intersections to preserve 3D information during rendering. Given the depth of a ray as $t$, GoF defines the ray-Gaussian intersection as the point $t^*$ where the Gaussian reaches its maximum value along the ray. Then the contribution $\varepsilon$ of the $k$-th Gaussian for a ray with camera center $o \in R^3$ and direction $r \in R^3$ is $\varepsilon(\mathcal{G}_k, o, r) = e^{-\frac{1}{2}(t^* - \mu_k)^T \Sigma_k^{-1} (t^* - \mu_k)}$. Similar to 3DGS, the image rendering is achieved by $\alpha$-blending according to the primitives' depth order $1, \ldots, K$ as:

$$c(o, r) = \sum_{k=1}^{K} c_k \cdot \alpha_k \cdot \varepsilon(\mathcal{G}_k, o, r) \cdot \prod_{j=1}^{k-1} \left(1 - \alpha_j \cdot \varepsilon(\mathcal{G}_j, o, r)\right). \qquad (2)$$

In this project, we render with GoF instead of 3DGS for its simplicity and superior performance.

**Linear Blend Skinning.** To realize rigid human deformations, a widely adopted paradigm is to define geometry and appearance in canonical space and then use the Linear Blend Skinning (LBS) to transform a human body to given poses. For a point $x_c$ in canonical space, LBS takes the local bone transformations $\mathbf{B} = \{B_b\}_{b=1}^{N_b}$ and computes its corresponding position $x_o$ in the posed space as:

$$x_o = LBS(x_c, \mathbf{w}, \mathbf{B}) = \sum_{b=1}^{N_b} w_b \cdot B_b \cdot x_c. \qquad (3)$$

Here $\mathbf{w} = \{w_b\}_{b=1}^{N_b}$ represents the skinning weight outputted by a neural field to avoid inverse skinning computation (Qian et al., 2024), where $w_b \in [0, 1]$ and $\sum_{b=1}^{N_b} w_b = 1$. We also set $N_b = 24$ and $B_b \in R^{4 \times 4}$ to encode the translation and rotation information.

## 3.2 HUMAN MOTION MODULE

Inspired by the previous methods (Weng et al., 2022; Qian et al., 2024; Song et al., 2025a), our method performs a non-rigid human deformation and a following rigid translation for the canonical 3D Gaussians. The non-rigid deformation aims to encode the pose-dependent fine-grained cloth

details, while the rigid counterpart wants to achieve large-scale human motions. Taking a Gaussian mean and a given pose as input, we formulate the non-rigid motion module as:

$$(\delta x, \delta s, \delta q, z) = \mathbb{MLP}_{nr}(x_c; \theta_p),\tag{4}$$

where $x_c$ is the mean of a canonical Gaussian and $(\delta x, \delta s, \delta q)$ represent the corresponding offsets of its position, scale and rotation. Additionally, $\theta_p$ is a latent code of the input SMPL pose while $\mathbb{MLP}_{nr}$ is a lightweight multilayer perceptron network. The feature vector $z$ is used to indicate local deformation when rendering images later. Then the canonical Gaussian is deformed as:

$$\mathbf{x}_d = \mathbf{x}_c + \delta x, \quad \mathbf{s}_d = \mathbf{s}_c \cdot \exp(\delta s), \quad \mathbf{q}_d = \mathbf{q}_c \cdot [1, \delta q_1, \delta q_2, \delta q_3].\tag{5}$$

Given the skinning weights $\{w_b\}_{b=1}^{N_b}$ and matched local bone transformations $\{B_b\}_{b=1}^{N_b}$, the non-rigidly deformed Gaussians are further transformed to the observation space as:

$$\mathbf{T} = \sum_{b=1}^{N_b} w_b \cdot B_b, \quad \mathbf{x}_o = \mathbf{T} \cdot \mathbf{x}_d, \quad \mathbf{R}_o = \mathbf{T}_{1:3,1:3} \cdot \mathbf{R}_d.\tag{6}$$

$\mathbf{R}_d$ here means the rotation matrix derived from the quaternion $\mathbf{q}_d$.

### 3.3 Motion-Aware Surface Smoothing

As discussed in Sec. 1, unconstrained geometry modeling prevents generalization to monocular inputs. To address this issue, we put smoothness constraints on the 3D Gaussian primitives to enhance their spatial distributions using a regularization term and a geometry-aware density control strategy.

Given the 3D Gaussians $\{\mathcal{G}^o\}$ in observation space, we render a depth map $I^d$ with input camera parameters. Specifically, the depth value of the $i$-th pixel is assigned by matching the nearest Gaussian in $\{\mathcal{G}^o\}$ along the ray, which is denoted as $\tilde{\mathcal{G}}_i^o$, and then computing the distance between the center of $\tilde{\mathcal{G}}_i^o$ and the camera center. The smoothness map $S$ is computed over the depth map $I^d$ as

$$S_i = \sum_j w_j \cdot |I_j^d - I_i^d|, \text{ with } w_j = \frac{e^{-\frac{\hat{d}_j^2}{2\sigma_s^2}}}{\sum_{j \in N(i)} e^{-\frac{\hat{d}_j^2}{2\sigma_s^2}}} \text{ and } \hat{d}_j = (\tilde{\mu}_j^c - \tilde{\mu}_i^c),\tag{7}$$

where $j$ is the nearby pixel indices $N(i)$ of the $i$th pixel, $\tilde{\mu}_i^c$ is the corresponding canonical Gaussian center of $\tilde{\mathcal{G}}_i^o$, and $\sigma_s$ is a hyper-parameter to adjust the sharpness of weight computation. To encourage the observed Gaussians $\{\mathcal{G}^o\}$ to distribute smoothly, we simply average over the smoothness map $S$ to regularize the network training as $\mathcal{L}_{smooth}^r = \frac{1}{n} \sum S_i$, where n is the total pixel number of $S$.

Compared to simply averaging the depth differences for each pixel of $S$ by setting $w_j = 1$, the smoothness weight $w_j$ can better serve for the articulation nature of humans by filtering out the nearby depth values caused by self-occlusions. The $w_j$ computation can further help preserve the geometric structures by only considering spatially neighbouring entities to avoid the negative effects of potential holes; please refer to Fig. 1 as an example.

As a self-supervised strategy, we can also render depth maps and compute smoothness regularization for arbitrary camera views, other than the input camera directions. Based on this, we define 32 fixed camera parameters, including positions and corresponding viewpoints, to cover the whole 3D space surrounding the human in videos. In each iteration, we randomly choose one virtual camera and compute the smoothness regularization as $\mathcal{L}_{smooth}^v$ to supervise motions under unseen views.

Besides acting as a regularization term, we additionally develop a geometry-aware Adaptive Density Control mechanism based on the smoothness computation. We observe that, by querying the smoothness signal for each Gaussian, we are able to identify Gaussian primitives that contribute less to smooth surfaces, allowing for more precise control over the placement and density of Gaussians. Specifically, we set the initial smoothness value to 100 for each Gaussian every K iterations. During each training iteration, the smoothness value $\eta$ of $\tilde{\mathcal{G}}_i^o$, which matches the $i$-th pixel of the smoothness map $S$, is updated as $\eta = \min\{\eta, S_i\}$. Similar to VolSDF (Yariv et al., 2021), we convert the smoothness value $\eta$ to a positive value as $\tau = exp(-\eta^2/(2\sigma_c^2))$, where $\sigma_c$ is a hyper-parameter. Then, for each Gaussian primitive with value $\tau$, its criteria can then be defined as

$$\xi_g = \nabla_g \cdot \tau, \quad \xi_p = \sigma_a \cdot (1 - \tau),\tag{8}$$

where $\nabla_g$ and $\sigma_a$ are the averaged gradient of Gaussian primitives and the aggregated opacity accumulated over K training iterations. New Gaussian primitives are added if $\xi_g$ is greater than a predefined threshold while Gaussians with $\xi_p$ less than a threshold are pruned. The smoothness-based density modulation helps produce Gaussians, contributing more to smooth surfaces.

## 3.4 NETWORK OPTIMIZATION

During training, we update learnable parameters of canonical 3D Gaussians $\{\mathcal{G}_c\}$, non-rigid deformation network and rigid skinning network in an end-to-end manner. Instead of storing spherical harmonics coefficients per Gaussian, we employ a learnable color network to better generalize to the monocular setting as in 3DGS-Avatar (Qian et al., 2024). For a frame with the viewing direction $\mathbf{d}$ and transformation matrix $\mathbf{T}$ as Eq. 6, we have $c = \mathbb{MLP}_{color}(f, z, \gamma(\bar{d}))$. Here $f \in R^3$ and z respectively denote the per-Gaussian color feature vector and the feature vector outputted by the non-rigid deformation network in Eq. 4. And $\gamma$ is the spherical harmonics basis function (Kerbl et al., 2023a) while $\bar{d}$ is obtained as $\bar{d} = \mathbf{T}_{1:3,1:3}^{-1} \cdot \mathbf{d}$ to canonicalize the viewing direction for better generalizations. As the input SMPL poses can be inaccurate, we additionally learn to update these parameters during training. In detail, we initialize the human poses with the SMLP parameters estimated from input images and then directly optimize these parameters via gradient back-propagation.

As pointed out by 2DGS (Huang et al., 2024), it is crucial to locally align Gaussian splats with the actual surfaces. To this end, we first define the normal of a Gaussian as the normal of its ray-Gaussian intersection plane given a ray direction (Yu et al., 2024b). Then we accumulate the normals of the 3D Gaussians using $\alpha$-blending along the ray with center $o$ and direction $r$ as $\mathbf{N} = \sum_i n_i \cdot \alpha_i \cdot \varepsilon(\mathcal{G}_i, o, r) \cdot \prod_{j=1}^{i-1}(1 - \alpha_j \cdot \varepsilon(\mathcal{G}_j, o, r))$, where $i$ indexes over intersected splats, $n_i$ is the normal of $i$-th Gaussian and $\mathbf{N}$ is the surface normal. To encourage normal consistency, we define a regularization term as $\mathcal{L}_{nc} = \sum_i (1 - n_i^T \mathbf{N})$. Inspired by existing work (Chen et al., 2024b), we leverage the normal maps from Sapiens (Khirodkar et al., 2024) to further constrain our estimated normal maps and have another normal regularization as $\mathcal{L}_{sap} = \|N - N_{gt}\|$, where $N_{gt}$ means the normal map from the Sapiens model. Finally, the total loss with weights $\{\lambda_{lpips}, \lambda_{skin}, \lambda_{smooth}, \lambda_{nc}, \lambda_{sap}\}$ for joint learning is defined as

$$\mathcal{L} = \mathcal{L}_{rgb} + \lambda_{lpips}\mathcal{L}_{lpips} + \lambda_{skin}\mathcal{L}_{skin} + \lambda_{smooth}\mathcal{L}_{smooth} + \lambda_{nc}\mathcal{L}_{nc} + \lambda_{sap}\mathcal{L}_{sap}, \quad (9)$$

where $\mathcal{L}_{smooth} = \mathcal{L}_{smooth}^r + \mathcal{L}_{smooth}^v$. See Sec. B for details.

## 4 RESULTS

We experimentally compare with latest human modeling methods, including Dyco (Chen et al., 2024c), 3DGS-Avatar (Qian et al., 2024), GauHuman (Hu et al., 2024b), GoMAvatar (Wen et al., 2024), LS-Avatar (Song et al., 2025a) and ToMiE (Zhan et al., 2025). Specifically, we evaluate across the tasks of novel view synthesis, novel pose animation and 3D shape estimation. We also assess the importance of each proposed technical contribution through ablation studies. In the following, we refer to "3DGS-Avatar" as "3DGS-A" for short.

Table 1: **Ablation study on MVHumanNet dataset.** Our full model outperforms all ablated baselines across most metrics, consistently proving the importance of all network components.

| | Novel View | | | Novel Pose | | |
|---|---|---|---|---|---|---|
| | PSNR↑ | SSIM↑ | LPIPS↓ | PSNR↑ | SSIM↑ | LPIPS↓ |
| w/o $\mathcal{L}_{nc}$ | 25.10 | 0.966 | 33.56 | 22.98 | 0.956 | 45.39 |
| w/o $\mathcal{L}_{smooth}^v$ | 24.80 | 0.967 | 32.86 | 22.70 | 0.956 | 45.81 |
| w/o GA-ADC | 25.17 | 0.966 | 34.55 | 23.00 | 0.956 | 46.59 |
| w/o $\{w_j\}$ | 24.93 | 0.965 | 35.47 | 23.01 | 0.955 | 47.20 |
| w/o $\mathcal{L}_{smooth}$ | 24.98 | 0.964 | 38.57 | **23.03** | 0.954 | 50.05 |
| w/o $\mathcal{L}_{sap}$ | 24.80 | 0.964 | 39.95 | 22.96 | 0.954 | 51.17 |
| only 3DGS | 24.73 | 0.963 | 40.51 | 22.95 | 0.954 | 51.19 |
| $w_j = 1$ | 24.71 | 0.965 | 34.35 | 22.81 | 0.955 | 46.69 |
| Canonical Smoothing | 25.16 | 0.966 | 34.48 | 22.98 | 0.956 | 46.58 |
| **Ours (full)** | **25.30** | **0.968** | **32.10** | 23.00 | **0.957** | **44.56** |

### 4.1 EXPERIMENTAL SETTINGS

Following ToMiE, we choose eight sequences from the DNA-Rendering dataset (Cheng et al., 2023) to evaluate results on loose-fitting clothing with complex textures. Similar to LS-Avatar, we also utilize eight sequences from the MVHumanNet dataset (Xiong et al., 2024) to further highlight the comparisons on complicated dressings and select four characters from ActorsHQ (Işık et al., 2023) for high-resolution image synthesis. We additionally adopt three sequences from MonoPerfCap (Xu et al., 2018) and Youtube videos (Weng et al., 2022; Yu et al., 2023) respectively as an in-the-wild

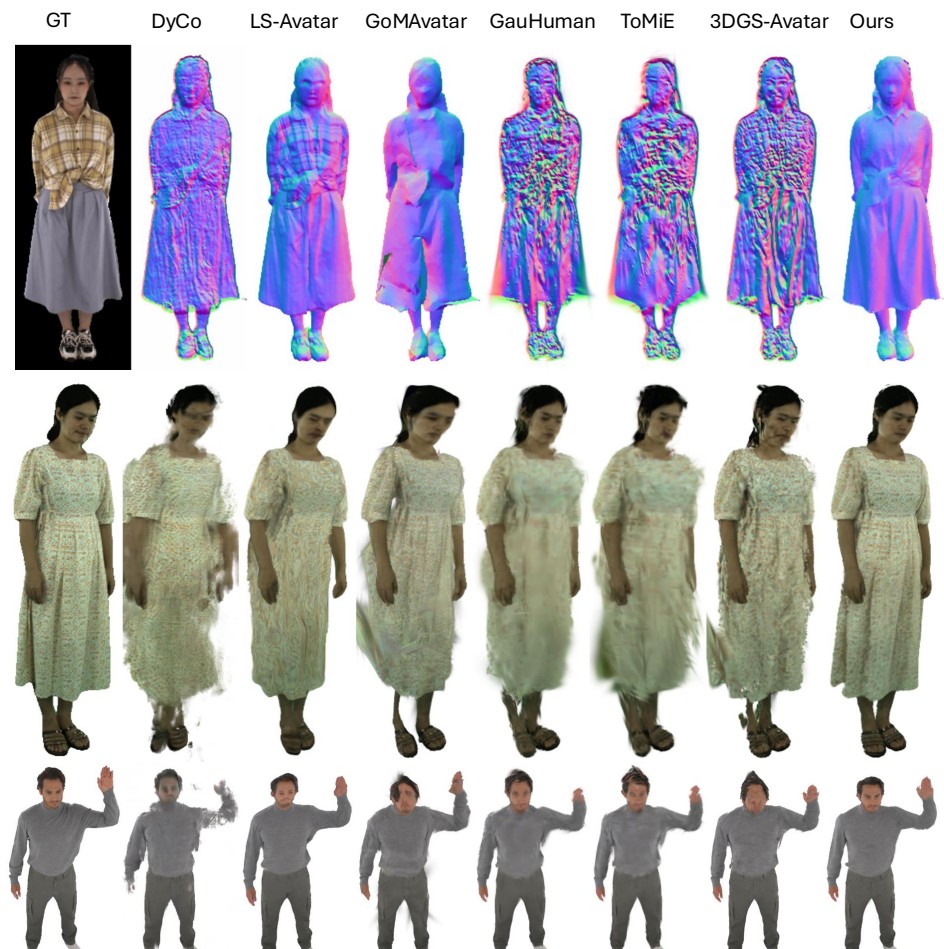

Figure 3: **Visual Comparisons with baselines.** Only our method can synthesize realistic normal map and output vivid texture patterns while baselines distort shape boundaries and introduce artifacts.

dataset. We employ the open-source SAM model (Kirillov et al., 2023) to segment accurate masks and obtain approximate camera and body poses with off-the-shelf estimators (Guo et al., 2023).

We report Peak Signal-to-Noise Ratio (PSNR) and Structural Similarity Index Measure (SSIM) to assess image quality in pixel space, and perceptual metrics like Learned Perceptual Image Patch Similarity (LPIPS) (Zhang et al., 2018), Fréchet Inception Distance (FID) (Heusel et al., 2017) and Kernel Inception Distance (KID) (Bińkowski et al., 2018) to compare structural accuracy and textured details in latent semantic space respectively. We multiply the LPIPS and KID scores by 1000 for better demonstrations. To quantitatively measure the geometry modeling, we compute the mean angular error and accuracy within $22.5°$ for surface normal estimation (Fu et al., 2024). We compute the ground truths for normal maps using the Sapiens model (Khirodkar et al., 2024).

## 4.2 COMPARISON TO THE STATE OF THE ART

Fig. 3 illustrates the qualitative comparisons with prior methods on the datasets of MVHumanNet, DNA-Rendering and ActorsHQ respectively. In comparison with baselines, our method can more faithfully synthesize the facial structure in both avatars and shape outlines. Additionally, we can show more realistic textured variations (e.g. wrinkles) in the whole human body, featuring our benefits in outputting high-frequency geometric patterns. In Tab. 2 and Tab. 3, we calculate PSNR, SSIM, LPIPS, KID and FID scores to quantitatively evaluate each method's image generation capability. Overall, our proposed approach produces best performance in most scenarios on PSNR and our method's LPIPS, KID and FID scores significantly improve on all baselines, further supporting our

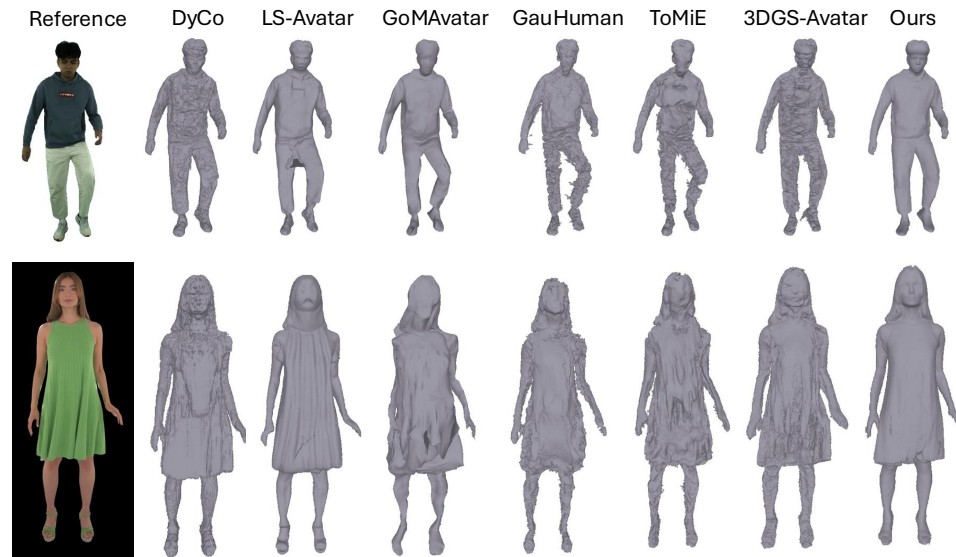

Figure 4: Mesh generation on the MVHumanNet and ActorsHQ datasets.

advantages. Note that the perceptual metrics (e.g. LPIPS) are reported to be more informative than the pixel-wise metrics like PSNR which are susceptible to slight output misalignment (Qian et al., 2024) and varying outdoor lighting conditions (Su et al., 2023).

The "Normal Estimation" columns in Tab. 2 and Tab. 3 further support aforementioned conclusions quantitatively. Specifically, our method advances the generation quality of normal maps by a large margin according to the Mean angular error (as Mean ↓) and accuracy within $22.5°$ (as $22.5°$ ↑). The per-object scores are listed in Tab. L - M for details. We exemplify the mesh generation in Fig. 4 to additionally reveal our geometry modeling capability. Our method produces sharper geometric structures and more adaptive fine-grained details than all baselines. In contrast, the baselines either introduce significant noise that distorts geometric patterns or smooth out prominent cloth wrinkles.

### 4.3 ABLATION STUDY

We ablate our important network components on the full MVHumanNet sequences to validate their contributions. Successively, we perform the following models by progressively removing components. Each model is built on top of the preceding one and we have: (1) Removing the normal consistency regularization $\mathcal{L}_{nc}$ as w/o $\mathcal{L}_{nc}$; (2) Removing effects of virtual cameras as w/o $\mathcal{L}_{smooth}^{v}$; (3) Removing the geometry-aware ADC as w/o GA-ADC; (4) Removing the computation of smoothness weight $\{w_j\}$ as w/o $\{w_j\}$; (5) Further removing $\mathcal{L}_{smooth}^{r}$ as w/o $\mathcal{L}_{smooth}$; (6) Removing $\mathcal{L}_{sap}$ as w/o $\mathcal{L}_{sap}$; (7) Replacing the GoF operator with 3DGS as only 3DGS. Fig. 5 demonstrates that removing the proposed components degrades network performance, leading to more artifacts and distorted shape boundaries. Tab. 1 and Sec. D further report the quantitative numbers and additional ablation results to conclude that all used technical components contribute to the optimal performance.

## 5 CONCLUSION

We introduce surface constraints for 3D Gaussian Splatting that enable accurate avatar modeling from monocular videos. The core idea is to explicitly quantify the smoothness of depth maps, which guides the compact and consistent distribution of 3D Gaussians. We demonstrate the effectiveness of our approach on six public datasets across three tasks, achieving superior image quality and more realistic human surface reconstruction compared to prior methods.

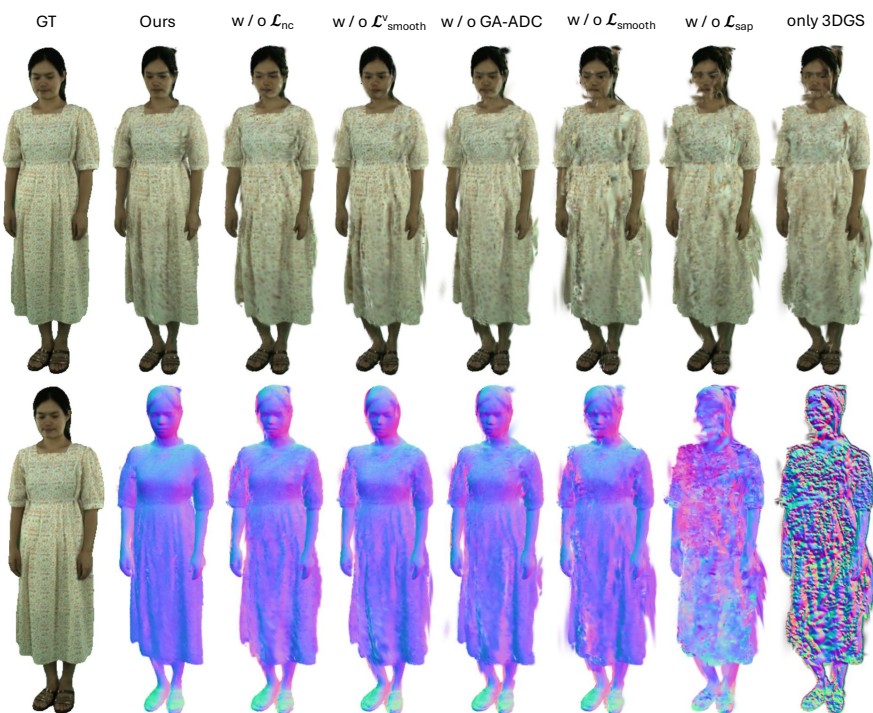

Figure 5: **Ablation Studies.** By successively ablating our proposed components, the networks degrade by yielding worse shape outlines and distorting local textures.

Table 2: **Quantitative comparisons on MVHumanNet dataset and outdoor sequences.** Our motion-aware surface smoothing outperforms all baselines across all four metrics, achieving over **10%** improvement in LPIPS, **10%** in FID, and **40%** in KID. The "Normal Estimation" column shows our strong modeling capability in geometry by synthesizing notably best normal maps among all methods. Cell color indicates best and second best for clearer comparisons.

| | Novel View Synthesis | | | | Novel Pose Animation | | | | Normal Estimation | | Outdoors (Novel Pose) | | |
|---|---|---|---|---|---|---|---|---|---|---|---|---|---|
| | PSNR↑ | LPIPS↓ | FID↓ | KID↓ | PSNR↑ | LPIPS↓ | FID↓ | KID↓ | Mean↓ | 22.5°↑ | PSNR↑ | SSIM↑ | LPIPS↓ |
| Dyco | 22.72 | 51.03 | 93.44 | 81.79 | 20.98 | 68.21 | 164.94 | 110.75 | 52.13 | 19.24 | 24.16 | 0.960 | 33.44 |
| LS-Avatar | 23.14 | 40.65 | 62.59 | 39.23 | 22.24 | 49.82 | 101.34 | 49.03 | 34.79 | 37.20 | 24.61 | 0.962 | 30.89 |
| GauHuman | 24.42 | 47.58 | 113.87 | 107.35 | 22.79 | 57.43 | 145.25 | 98.26 | 57.52 | 16.77 | 25.06 | 0.964 | 37.47 |
| GoMAvatar | 23.73 | 42.25 | 80.89 | 55.91 | 22.26 | 53.85 | 119.78 | 55.73 | 56.75 | 4.37 | 24.43 | 0.961 | 31.54 |
| ToMie | 24.20 | 51.81 | 126.93 | 117.56 | 22.53 | 62.09 | 158.92 | 109.00 | 60.52 | 13.93 | 25.06 | 0.964 | 37.75 |
| 3DGS-Avatar | 24.39 | 41.25 | 85.69 | 71.17 | 22.81 | 51.47 | 118.64 | 63.75 | 59.33 | 13.53 | 24.90 | 0.964 | 29.52 |
| Ours | 25.30 | 32.10 | 50.71 | 26.11 | 23.00 | 44.56 | 92.02 | 31.17 | 22.20 | 66.95 | 25.29 | 0.966 | 29.00 |

Table 3: **Quantitative comparisons on the datasets of DNA-Rendering and ActorsHQ.** Our method can consistently improve all baselines across all metrics in the DNA-Rendering dataset, especially in LPIPS. As discussed in Sec. E, although we can achieve comparable or even better results than baselines on ActorsHQ dataset, how to recreate realistic results for high-resolution long sequences needs further exploration.

| | DNA (Novel View) | | | DNA (Novel Pose) | | | Normal Estimation | | ActorsHQ (Novel View) | | | ActorsHQ (Novel Pose) | | |
|---|---|---|---|---|---|---|---|---|---|---|---|---|---|---|
| | PSNR↑ | SSIM↑ | LPIPS↓ | PSNR↑ | SSIM↑ | LPIPS↓ | Mean↓ | 22.5°↑ | PSNR↑ | SSIM↑ | LPIPS↓ | PSNR↑ | SSIM↑ | LPIPS↓ |
| Dyco | 26.10 | 0.949 | 54.55 | 23.73 | 0.929 | 79.14 | 53.32 | 20.42 | 20.91 | 0.936 | 126.02 | 19.50 | 0.927 | 138.43 |
| LS-Avatar | 26.53 | 0.954 | 42.76 | 24.36 | 0.943 | 59.79 | 43.63 | 33.48 | 21.30 | 0.940 | 98.55 | 20.34 | 0.936 | 103.27 |
| GauHuman | 27.25 | 0.956 | 56.82 | 24.83 | 0.941 | 71.91 | 66.36 | 12.78 | 22.13 | 0.944 | 114.43 | 20.53 | 0.937 | 118.80 |
| GoMAvatar | 27.10 | 0.958 | 42.36 | 24.16 | 0.938 | 64.26 | 55.32 | 5.81 | 20.42 | 0.933 | 118.57 | 19.40 | 0.928 | 123.95 |
| ToMie | 27.56 | 0.957 | 58.24 | 24.82 | 0.940 | 74.55 | 65.25 | 13.52 | 22.15 | 0.943 | 117.01 | 20.71 | 0.937 | 119.79 |
| 3DGS-Avatar | 27.85 | 0.960 | 42.84 | 24.69 | 0.940 | 63.46 | 58.65 | 15.45 | 21.47 | 0.939 | 101.05 | 20.23 | 0.934 | 106.65 |
| Ours | 28.78 | 0.967 | 35.23 | 25.00 | 0.946 | 55.80 | 20.46 | 72.66 | 22.31 | 0.948 | 89.08 | 20.28 | 0.936 | 100.54 |

# 6 REPRODUCIBILITY STATEMENT

To ensure reproducibility, we present our method in Sec. 3, including both the high-level motivations and the corresponding design choices. Additional implementation details are provided in Sec. B to complement the methodological explanations at a lower level. Finally, we will release our code to enable full reproducibility of the proposed framework.

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

# Appendices

In the appendix, we first provide an extended discussion of related work. We then describe the implementation details of our framework and the datasets used for evaluation. Additional experimental examples are included to support our technical motivations, followed by comprehensive quantitative comparisons across multiple datasets and evaluation metrics. Finally, we discuss the limitations and potential social impacts of this work, and include a statement on the use of LLMs. The attached video showcases the animation results.

## A    More Discussions on Related Works

**More on Avatar Representations.** The pioneering SMPL model (Loper et al., 2015; Pavlakos et al., 2019) provides a parametric representation of human body shape and has served as a cornerstone for many recent neural animatable avatar frameworks. Beyond the works discussed in Sec. 2, several early efforts have explored NeRF-based representations for avatar learning (Kwon et al., 2021; Peng et al., 2021b; Gafni et al., 2021; Goel et al., 2023; Geng et al., 2023; Chen et al., 2023b; Jiang et al., 2023; Wang et al., 2024a). More recently, advances leveraging 3D Gaussian Splatting (3DGS) have enabled faster and more accurate avatar rendering (Hu et al., 2024a; Jena et al., 2023; Li et al., 2023a; 2024; Pang et al., 2024; Zheng et al., 2024a;b). To address pose ambiguities, works such as (Chen et al., 2024c; Hu et al., 2024c) explicitly incorporate motion context from input pose sequences. Meanwhile, diffusion-based approaches (Hu, 2024; Men et al., 2025) have been proposed to improve rendering realism but remain constrained by the traditional SMPL formulation. In contrast, our contribution lies in introducing a simple yet effective surface smoothness prior for articulated avatars, which substantially enhances both rendering fidelity and geometric accuracy.

**Neural Rendering.** In recent years, the computer graphics and 3D vision communities (Park et al., 2019; Wu et al., 2019; Mescheder et al., 2019; Takikawa et al., 2021; Müller et al., 2022; Zhang et al., 2024a) have devoted significant attention to neural fields, owing to their continuous representations and ability to generate outputs at arbitrary resolutions. Building on this idea, Neural Radiance Fields (NeRF) (Mildenhall et al., 2020) transformed photorealistic rendering by capturing view-dependent effects. NeRF and its variants learn mappings from 3D query points to color and density values, which are then accumulated into pixel intensities via volumetric rendering. Since its introduction, extensive research has sought to reduce computational overhead (Chen et al., 2022; Müller et al., 2022; Fridovich-Keil et al., 2022), extend modeling to dynamic scenes (Li et al., 2021b; Park et al., 2021; Pumarola et al., 2021), and enhance geometric fidelity (Wang et al., 2021; Yariv et al., 2021).

More recently, the 3D Gaussian Splatting (3DGS) framework (Kerbl et al., 2023a) has proposed representing scenes with anisotropic Gaussians and rasterizing them as geometric primitives, achieving real-time high-quality rendering. Extensions such as Scaffold-GS (Lu et al., 2024) further improve image quality and memory efficiency through hierarchical structures. To handle dynamic scenes, subsequent works extend 3DGS by learning time-dependent Gaussian parameters (Luiten et al., 2024; Wu et al., 2024a; Kerbl et al., 2023b) or by decomposing motion into bases and coefficients (Wang et al., 2025). However, these approaches primarily emphasize novel view synthesis rather than accurate surface reconstruction. Additionally, our work aims to advance the generalization of animatable avatars from monocular videos by introducing a simple yet effective geometric prior.

## B    Implementation Details

Our method is implemented in the PyTorch framework (Paszke et al., 2019), and optimized with Adam (Kingma & Ba, 2014) using the default settings $\beta_1 = 0.9$ and $\beta_2 = 0.999$. The learning rate of 3D Gaussians follows the original 3DGS setting (Kerbl et al., 2023a), and a step-decay schedule is applied for stable convergence. All MLP weights are activated with ReLU (Agarap, 2018) for training stability.

We initialize the canonical 3D Gaussians with 50K random points sampled from the SMPL template in T-pose. The skinning network is a four-layer MLP ($128 \times 128$ each) that predicts LBS weights. For non-rigid deformations, the input 3D positions are first encoded with OneBlob (Müller et al.,

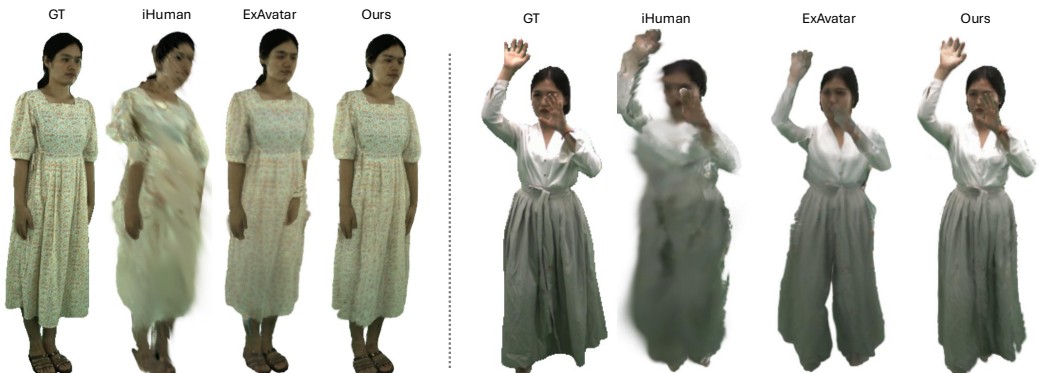

Figure A: **Visual Comparisons with additional baselines.** Our method produces the most coherent and continuous shape outlines with adaptive wrinkle details, whereas the baselines tend to distort shape structures or blur geometric features.

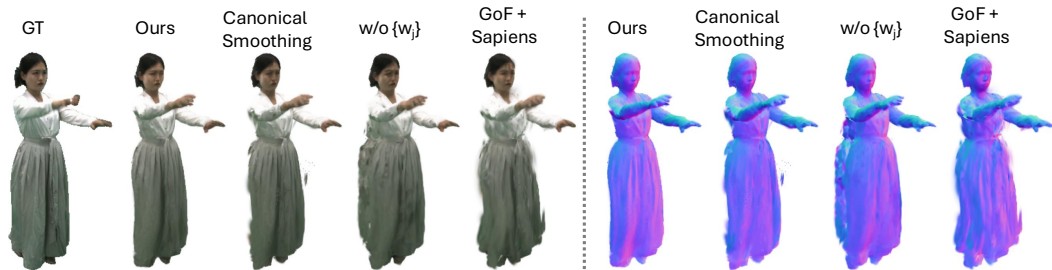

Figure B: **Additional Ablation Studies.** Only our full model equipped with motion-aware surface smoothing can reasonably reconstruct the RGB rendering and normal map estimations.

2019), concatenated with a pose latent code, and then processed by a three-layer MLP ($128 \times 128$ each) to predict offsets in position, rotation, and scale.

We define the skinning weight regularization loss following 3DGS-Avatar. To regularize training and reduce overfitting, we perturb both human poses and viewing directions. Specifically, SMPL pose parameters are perturbed with Gaussian noise $\mathcal{N}(0, 0.1)$ with probability $p = 0.5$, while input viewing directions are perturbed with uniform noise after canonicalization via inverse skinning. Following HumanNeRF (Weng et al., 2022) and 3DGS-Avatar, we also adopt a delayed optimization strategy. During the first 1K iterations, only the skinning network is optimized; 3D Gaussian updates begin afterward; the non-rigid deformation network is trained after 3K iterations; and the pose correction module is enabled after 5K iterations.

For supervision, rendered patches and their ground-truth counterparts are embedded into the VGG feature space (Simonyan & Zisserman, 2014) to compute LPIPS metrics. Offline pose estimates are further refined during training to alleviate pose errors.

To ensure consistent comparisons and fair evaluation of generalization, we keep the same hyperparameter settings across all experiments. This includes the weights of the loss function $\lambda_{lpips}, \lambda_{skin}, \lambda_{smooth}, \lambda_{nc}, \lambda_{sap}$, the number of training iterations, and the learning rate schedule. These values are chosen based on empirical performance on standard benchmarks. We train our model on a single NVIDIA RTX 3090 GPU, with each run requiring roughly one hour.

## C  DATASET DETAILS

**DNA-Rendering Dataset (Cheng et al., 2023).** Following the experimental settings of ToMie, we select eight subjects (0007_04, 0010_03, 0017_11, 0025_09, 0042_12, 0044_10, 0124_03, 0813_05) wearing loose-fitting clothing with complex textures to evaluate our human modeling capability. For each subject, the first 100 frames are used for training, while the remaining frames are reserved

Table A: **Novel-view synthesis comparisons on MVHumanNet** with PSNR and LPIPS scores. Cell color indicates best .

| | Dyco | | LS-Avatar | | GauHuman | | GoMAvatar | | ToMie | | 3DGS-Avatar | | **Ours** | |
|---|---|---|---|---|---|---|---|---|---|---|---|---|---|---|
| | PSNR ↑ | LPIPS ↓ | PSNR ↑ | LPIPS ↓ | PSNR ↑ | LPIPS ↓ | PSNR ↑ | LPIPS ↓ | PSNR ↑ | LPIPS ↓ | PSNR ↑ | LPIPS ↓ | PSNR ↑ | LPIPS ↓ |
| 100846 | 21.08 | 63.78 | 23.21 | 45.13 | 24.36 | 50.29 | 23.32 | 45.00 | 24.48 | 52.54 | 24.52 | 41.99 | 25.68 | 31.81 |
| 100990 | 22.81 | 44.51 | 23.39 | 37.20 | 23.84 | 45.02 | 22.66 | 44.69 | 22.86 | 52.56 | 23.87 | 40.78 | 25.10 | 27.60 |
| 102107 | 23.12 | 51.75 | 22.96 | 44.87 | 24.28 | 55.14 | 23.36 | 48.72 | 24.11 | 60.69 | 24.20 | 49.75 | 24.76 | 36.58 |
| 102145 | 24.01 | 41.09 | 24.85 | 35.34 | 25.54 | 42.13 | 24.89 | 38.41 | 25.73 | 44.73 | 25.72 | 35.42 | 26.63 | 28.15 |
| 103708 | 21.50 | 48.09 | 23.55 | 34.19 | 25.22 | 37.58 | 24.14 | 35.35 | 25.10 | 39.65 | 24.77 | 34.92 | 24.85 | 27.30 |
| 200173 | 21.90 | 69.73 | 22.56 | 50.42 | 23.54 | 64.57 | 22.94 | 57.24 | 22.95 | 72.12 | 23.13 | 54.11 | 23.88 | 41.58 |
| 204112 | 24.42 | 36.99 | 24.07 | 32.22 | 24.67 | 38.78 | 24.87 | 29.99 | 24.65 | 41.29 | 24.62 | 31.49 | 25.02 | 28.62 |
| 204129 | 22.94 | 52.32 | 20.56 | 45.83 | 23.94 | 47.11 | 23.64 | 38.57 | 23.74 | 50.90 | 24.34 | 41.50 | 25.45 | 30.32 |
| Avg | 22.72 | 51.03 | 23.14 | 40.65 | 24.42 | 47.57 | 23.73 | 42.25 | 24.20 | 51.81 | 24.39 | 41.25 | 25.30 | 32.10 |

Table B: Novel-view synthesis comparisons with two additional baselines on MVHumanNet with PSNR and LPIPS scores. Cell color indicates best .

| | 100846 | | 100990 | | 102107 | | 102145 | | 103708 | | 200173 | | 204112 | | 204129 | | Avg | |
|---|---|---|---|---|---|---|---|---|---|---|---|---|---|---|---|---|---|---|
| | PSNR ↑ | LPIPS ↓ | PSNR ↑ | LPIPS ↓ | PSNR ↑ | LPIPS ↓ | PSNR ↑ | LPIPS ↓ | PSNR ↑ | LPIPS ↓ | PSNR ↑ | LPIPS ↓ | PSNR ↑ | LPIPS ↓ | PSNR ↑ | LPIPS ↓ | PSNR ↑ | LPIPS ↓ |
| iHuman | 22.93 | 61.91 | 21.69 | 55.52 | 21.04 | 72.10 | 24.13 | 53.34 | 23.05 | 48.36 | 19.85 | 86.28 | 23.47 | 50.54 | 20.53 | 65.88 | 22.09 | 61.74 |
| ExAvatar | 23.66 | 39.97 | 23.57 | 37.15 | 23.03 | 44.02 | 24.20 | 37.26 | 24.80 | 31.93 | 21.46 | 57.61 | 23.66 | 35.55 | 24.18 | 37.04 | 23.57 | 40.07 |
| Ours | 25.68 | 31.81 | 25.10 | 27.60 | 24.76 | 36.58 | 26.63 | 28.15 | 24.85 | 27.30 | 23.88 | 41.58 | 25.02 | 28.62 | 25.45 | 30.32 | 25.30 | 32.10 |

for testing novel pose animation. We use "camera 26" for training and two surrounding cameras ("camera 23" and "camera 25") for novel view synthesis and unseen pose evaluation.

**MVHumanNet Dataset (Xiong et al., 2024).** To further assess generation quality for loose clothing, we select eight sequences (100846, 100990, 102107, 102145, 103708, 200173, 204112, 204129). As with the DNA-Rendering dataset, one camera view is used for training, while two neighboring views are used for evaluation.

**ActorHQ Dataset (Işık et al., 2023).** We choose four characters for our experiments: two females with loose dresses, one female with loose-fitting pants, and one male with tighter clothing. Following the LS-Avatar's evaluation protocol, we use 300 images from "camera 127" with indices in the range [500, 800) for training. For novel view synthesis, images from "camera 6", "camera 7", and "camera 8" are used. To evaluate novel pose animation, we further sample 100 images in the range [800, 900) from the same three cameras.

**YouTube Online Videos.** Starting with HumanNeRF (Weng et al., 2022), YouTube sequences have been widely adopted to test generalization to in-the-wild monocular videos (Yu et al., 2023; Wen et al., 2024; Song et al., 2025a). In this work, we evaluate on the "Way2sexy", "Story", and "Invisible" sequences downloaded from the internet. These monocular videos contain diverse human motions, enabling quantitative comparisons on the novel pose animation task.

**MonoPerfCap Dataset (Xu et al., 2018).** This dataset provides in-the-wild human performance sequences with complex real-world environments and diverse actions, along with high-resolution images and ground-truth masks. Following LS-Avatar, we use three video sequences from MonoPerfCap to evaluate robustness to unseen poses in monocular videos.

# D  MORE RESULTS

**Additional Ablation Studies.** To further highlight the importance of the proposed motion-aware surface smoothing, we perform the visual comparisons with three ablated variants. Specifically, we ablate our full model by removing the weight $w_j$ and computing the smoothness map for canonical Gaussians as the "Canonical Smoothing" to reveal the importance of applying surface smoothing in observation space rather than in canonical space. The "GoF + Sapiens" model represents the network only utilizing GoF operator for rendering and Sapiens' normal map as supervision while the w/o $\{w_j\}$ model is described in Sec. 4.3. As illustrated in Fig. B, only our full model can reasonably estimate the RGB rendering and normal map results.

Table C: **Novel-pose animation comparisons on MVHumanNet** with PSNR and LPIPS scores. Cell color indicates best .

|  | Dyco | | LS-Avatar | | GauHuman | | GoMAvatar | | ToMie | | 3DGS-Avatar | | **Ours** | |
|---|---|---|---|---|---|---|---|---|---|---|---|---|---|---|
|  | PSNR ↑ | LPIPS ↓ | PSNR ↑ | LPIPS ↓ | PSNR ↑ | LPIPS ↓ | PSNR ↑ | LPIPS ↓ | PSNR ↑ | LPIPS ↓ | PSNR ↑ | LPIPS ↓ | PSNR ↑ | LPIPS ↓ |
| 100846 | 19.78 | 80.90 | 22.52 | 50.07 | 22.66 | 56.08 | 22.19 | 53.50 | 22.57 | 57.83 | 23.27 | 48.53 | 23.70 | 42.05 |
| 100990 | 21.74 | 53.59 | 22.96 | 38.92 | 22.80 | 47.95 | 22.14 | 47.33 | 21.88 | 56.07 | 23.26 | 43.05 | 24.22 | 32.22 |
| 102107 | 21.92 | 65.37 | 22.91 | 49.59 | 23.47 | 62.15 | 22.64 | 56.98 | 23.51 | 68.86 | 23.24 | 57.70 | 22.96 | 47.10 |
| 102145 | 22.02 | 57.39 | 24.13 | 42.55 | 24.12 | 50.72 | 23.66 | 49.11 | 24.03 | 53.55 | 24.08 | 45.41 | 23.97 | 41.19 |
| 103708 | 20.39 | 58.60 | 22.42 | 42.78 | 23.49 | 45.90 | 22.44 | 44.91 | 23.51 | 48.59 | 23.17 | 43.03 | 23.37 | 35.91 |
| 200173 | 18.96 | 112.21 | 19.87 | 78.63 | 20.51 | 90.82 | 19.91 | 85.58 | 19.96 | 99.76 | 20.07 | 81.20 | 20.02 | 73.21 |
| 204112 | 21.79 | 50.18 | 22.91 | 38.25 | 22.98 | 44.91 | 23.05 | 37.75 | 22.68 | 47.65 | 22.95 | 37.51 | 23.06 | 36.66 |
| 204129 | 21.31 | 67.48 | 20.24 | 57.80 | 22.35 | 60.96 | 22.12 | 55.61 | 22.12 | 64.39 | 22.48 | 55.32 | 22.73 | 48.15 |
| Avg | 20.99 | 68.21 | 22.24 | 49.82 | 22.80 | 57.44 | 22.27 | 53.85 | 22.53 | 62.09 | 22.82 | 51.47 | 23.00 | 44.56 |

Table D: **Novel-pose animation comparisons with two additional baselines on MVHumanNet with PSNR and LPIPS scores. Cell color indicates best .**

|  | 100846 | | 100990 | | 102107 | | 102145 | | 103708 | | 200173 | | 204112 | | 204129 | | Avg | |
|---|---|---|---|---|---|---|---|---|---|---|---|---|---|---|---|---|---|---|
|  | PSNR ↑ | LPIPS ↓ | PSNR ↑ | LPIPS ↓ | PSNR ↑ | LPIPS ↓ | PSNR ↑ | LPIPS ↓ | PSNR ↑ | LPIPS ↓ | PSNR ↑ | LPIPS ↓ | PSNR ↑ | LPIPS ↓ | PSNR ↑ | LPIPS ↓ | PSNR ↑ | LPIPS ↓ |
| iHuman | 22.33 | 62.99 | 21.45 | 56.35 | 20.70 | 77.17 | 23.30 | 57.83 | 22.12 | 55.18 | 18.49 | 107.72 | 22.70 | 53.30 | 19.63 | 78.75 | 21.34 | 68.66 |
| ExAvatar | 23.23 | 42.66 | 19.89 | 56.15 | 21.25 | 54.94 | 23.68 | 42.64 | 23.29 | 37.66 | 19.52 | 77.68 | 22.48 | 39.79 | 22.56 | 47.85 | 21.98 | 49.92 |
| Ours | 23.70 | 42.05 | 24.22 | 32.22 | 22.96 | 47.10 | 23.97 | 41.19 | 23.37 | 35.91 | 20.02 | 73.21 | 23.06 | 36.66 | 22.73 | 48.15 | 23.00 | 44.56 |

**Comparisons with iHuman and ExAvatar.** To provide a more comprehensive comparison and draw a stronger conclusion, we include two additional baselines in Fig. A. Consistent with the conclusions in the main paper, our method outperforms these two baselines by generating more coherent shape boundaries and capturing adaptive geometric details. The per-sequence metric details are provided in Tab. B and Tab. D. Both the qualitative and quantitative results highlight the empirical advantages of our method over existing approaches.

**Detailed Quantative Comparisons.** Here we provide detailed score comparisons for image renderings and shape reconstructions. Specifically, Tab. A - K list the PSNR, LPIPS, FID and KID scores for the image rendering results on the MVHumanNet and DNA-Rendering datasets. Additionally, Tab. L and Tab. M provide the results for the normal estimation task to make a comprehensive evaluation. Finally, Tab. F looks into geometry reconstruction comparisons on the MVHumanNet dataset with Chamfer Distance (CD) and Normal Consistency (NC) to include mesh-level evaluations. Consistent with the observations in Fig. 4, our model consistently improves upon the baselines across nearly all sequences for both metrics.

**More Visual Comparisons.** Besides the image results shown in the main text, we visualize more rendering comparisons in Fig. C - Fig. K, corresponding geometry reconstruction in Fig. I. Consistent with the findings in the main text, our method maintains coarse body shapes and finely matched textures simultaneously for both novel view synthesis and unseen pose animation.

**Summary of Supplementary Video.** The supplementary video presents comparisons on the DNA-Rendering, MVHumanNet, and ActorsHQ datasets, as well as an outdoor sequence. For the outdoor video, we render novel poses, while for the other datasets we evaluate under novel view synthesis. Except for the YouTube sequence, human poses are taken directly from the official datasets. In the YouTube case, we adopt poses from AIST++ (Li et al., 2021a) to animate our avatar representations, demonstrating robustness to in-the-wild videos. We also provide normal map comparisons to highlight our ability to synthesize fine-grained geometric details.

# E    DISCUSSIONS & FUTURE WORKS

Our method achieves state-of-the-art or comparable performance with existing approaches in both image rendering and 3D human surface reconstruction. Nevertheless, the architecture is built upon the 3DGS framework, which requires re-training when applied to new input videos. How to generalize to arbitrary videos without retraining from scratch remains an intriguing open problem. A promising direction is to infer accurate human representations in a feed-forward manner using geometric

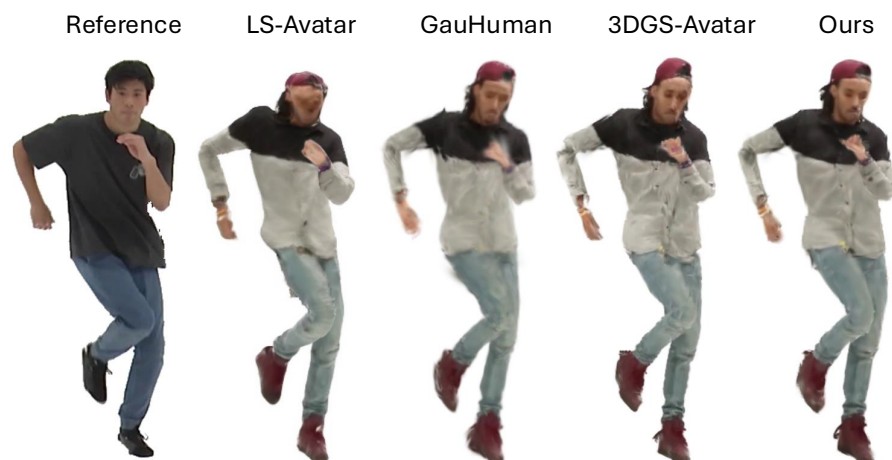

Reference     LS-Avatar     GauHuman     3DGS-Avatar     Ours

Figure C: **Motion retargeting on Youtube video.** Given the out-of-distribution pose driven from AIST++, only our method can accurately preserve the shape contours and fine-grained details (e.g. the cloth buttons). In contrast, baselines either severely blur the facial structures or distort the continuous geometry boundaries.

Table E: **Novel-pose animation comparisons on MVHumanNet** with FID and KID scores. Cell color indicates best.

|        | Dyco FID↓ | Dyco KID↓ | LS-Avatar FID↓ | LS-Avatar KID↓ | GauHuman FID↓ | GauHuman KID↓ | GoMAvatar FID↓ | GoMAvatar KID↓ | ToMie FID↓ | ToMie KID↓ | 3DGS-Avatar FID↓ | 3DGS-Avatar KID↓ | Ours FID↓ | Ours KID↓ |
|--------|------|------|------|------|------|------|------|------|------|------|------|------|------|------|
| 100846 | 333.58 | 257.07 | 170.09 | 123.09 | 222.75 | 181.64 | 181.66 | 101.74 | 261.14 | 212.20 | 177.76 | 103.71 | 109.85 | 34.35 |
| 100990 | 99.24 | 34.15 | 77.10 | 24.80 | 106.46 | 51.40 | 121.65 | 65.45 | 151.40 | 93.31 | 109.98 | 51.23 | 70.01 | 16.69 |
| 102107 | 103.49 | 51.89 | 64.82 | 15.78 | 122.98 | 78.49 | 82.53 | 28.65 | 141.97 | 95.84 | 100.25 | 63.39 | 65.38 | 17.53 |
| 102145 | 145.04 | 56.29 | 89.32 | 22.40 | 146.69 | 75.43 | 126.15 | 41.49 | 154.40 | 82.53 | 108.06 | 28.41 | 94.14 | 22.31 |
| 103708 | 203.93 | 212.90 | 151.47 | 117.19 | 156.23 | 126.59 | 137.42 | 81.03 | 166.36 | 139.84 | 150.29 | 127.92 | 102.10 | 45.15 |
| 200173 | 224.96 | 169.72 | 103.01 | 29.66 | 167.81 | 102.03 | 134.04 | 61.58 | 169.66 | 112.67 | 132.67 | 65.43 | 114.52 | 42.90 |
| 204112 | 76.15 | 24.37 | 50.83 | 4.51 | 76.21 | 34.32 | 57.26 | 9.43 | 85.06 | 43.21 | 57.40 | 10.14 | 57.38 | 7.17 |
| 204129 | 133.13 | 79.64 | 104.06 | 54.79 | 162.79 | 136.25 | 117.53 | 56.46 | 141.38 | 92.41 | 112.71 | 59.81 | 122.85 | 63.28 |
| Avg | 164.94 | 110.76 | 101.34 | 49.03 | 145.24 | 98.27 | 119.78 | 55.73 | 158.92 | 109.00 | 118.64 | 63.75 | 92.03 | 31.17 |

Table F: **Geometry reconstruction comparisons on MVHumanNet**. Following LS-Avatar (Song et al., 2025a), we compute the L2 Chamfer Distance (CD) and Normal Consistency (NC) for geometry evaluations. Cell color indicates best.

|        | DyCo CD↓ | DyCo NC↑ | LS-Avatar CD↓ | LS-Avatar NC↑ | GauHuman CD↓ | GauHuman NC↑ | GoM-Avatar CD↓ | GoM-Avatar NC↑ | ToMie CD↓ | ToMie NC↑ | 3DGS-Avatar CD↓ | 3DGS-Avatar NC↑ | Ours CD↓ | Ours NC↑ |
|--------|------|------|------|------|------|------|------|------|------|------|------|------|------|------|
| 100846 | 7.54 | 0.690 | 6.53 | 0.755 | 14.10 | 0.652 | 7.69 | 0.764 | 12.99 | 0.667 | 14.75 | 0.605 | 4.82 | 0.796 |
| 100990 | 6.34 | 0.718 | 2.94 | 0.806 | 7.76 | 0.662 | 3.19 | 0.810 | 7.58 | 0.681 | 6.78 | 0.698 | 2.46 | 0.827 |
| 102107 | 3.37 | 0.722 | 2.75 | 0.826 | 15.58 | 0.661 | 3.12 | 0.800 | 8.05 | 0.680 | 6.48 | 0.685 | 2.46 | 0.830 |
| 102145 | 6.94 | 0.692 | 5.85 | 0.785 | 23.65 | 0.669 | 12.95 | 0.753 | 16.10 | 0.711 | 19.37 | 0.636 | 3.31 | 0.809 |
| 103708 | 5.45 | 0.703 | 5.45 | 0.773 | 10.27 | 0.694 | 7.76 | 0.762 | 10.95 | 0.683 | 13.25 | 0.664 | 3.73 | 0.811 |
| 200173 | 10.99 | 0.654 | 9.84 | 0.751 | 22.43 | 0.650 | 12.81 | 0.698 | 19.03 | 0.635 | 15.86 | 0.603 | 7.48 | 0.740 |
| 204112 | 3.19 | 0.728 | 3.58 | 0.808 | 7.34 | 0.689 | 4.66 | 0.806 | 8.12 | 0.696 | 7.63 | 0.694 | 3.24 | 0.811 |
| 204129 | 4.29 | 0.749 | 4.46 | 0.781 | 12.20 | 0.626 | 6.18 | 0.751 | 10.87 | 0.671 | 11.90 | 0.625 | 5.62 | 0.793 |
| Avg | 6.02 | 0.707 | 5.18 | 0.786 | 14.17 | 0.663 | 7.30 | 0.768 | 11.71 | 0.678 | 12.00 | 0.651 | 4.14 | 0.802 |

foundation models (Wang et al., 2024b). In turn, it could also be possible to train foundational models on avatar reconstructions learned from monocular video as in this paper.

To further improve efficiency, another avenue is to incorporate grid-based methods (Müller et al., 2022; Chen et al., 2022; Wu et al., 2023; Lu et al., 2024) into dynamic 3DGS learning. As with

Table G: Comparisons on the training time, rendering speed and learnable parameter numbers.

| | iHuman | ExAvatar | GauHuman | GoM-Avatar | Dyco | 3DGS-Avatar | LS-Avatar | ToMie | **Ours** |
|---|---|---|---|---|---|---|---|---|---|
| Training Time | 3min | 3h | 25 min | 2d | 12h | 30 min | 15h | 30 min | 1h |
| Rendering Speed (FPS) | 70+ | 20 | 180+ | 1.2 | 0.2 | 80+ | 0.05 | 60+ | 60+ |
| Parameter Number (M) | 14.33 | 4.14 | 1.1 | 0.95 | 97.49 | 1.91 | 1.52 | 1.22 | 1.7 |

Table H: **Novel-view synthesis comparisons on DNA-Rendering** with PSNR and LPIPS scores. Cell color indicates best .

| | Dyco | | LS-Avatar | | GauHuman | | GoMAvatar | | ToMie | | 3DGS-Avatar | | **Ours** | |
|---|---|---|---|---|---|---|---|---|---|---|---|---|---|---|
| | PSNR ↑ | LPIPS ↓ | PSNR ↑ | LPIPS ↓ | PSNR ↑ | LPIPS ↓ | PSNR ↑ | LPIPS ↓ | PSNR ↑ | LPIPS ↓ | PSNR ↑ | LPIPS ↓ | PSNR ↑ | LPIPS ↓ |
| 0007_04 | 23.50 | 88.78 | 23.75 | 77.47 | 25.10 | 100.24 | 24.47 | 75.50 | 25.36 | 102.19 | 25.29 | 72.06 | 25.60 | 62.79 |
| 0010_03 | 25.34 | 43.71 | 24.61 | 44.73 | 25.46 | 56.49 | 25.52 | 41.96 | 25.59 | 58.08 | 26.11 | 40.89 | 26.60 | 38.01 |
| 0017_11 | 27.95 | 31.61 | 26.75 | 27.07 | 27.44 | 35.11 | 26.85 | 28.61 | 27.87 | 37.43 | 28.40 | 27.17 | 29.72 | 22.61 |
| 0025_09 | 27.52 | 45.92 | 27.66 | 39.74 | 28.23 | 60.09 | 27.69 | 43.37 | 28.83 | 56.71 | 28.75 | 42.88 | 30.55 | 33.58 |
| 0042_12 | 26.08 | 39.29 | 24.54 | 41.24 | 25.41 | 47.86 | 25.41 | 37.35 | 25.55 | 50.74 | 26.27 | 43.74 | 27.25 | 28.58 |
| 0044_10 | 28.56 | 27.21 | 27.36 | 28.98 | 28.46 | 42.94 | 28.75 | 27.69 | 28.61 | 41.97 | 28.56 | 28.56 | 29.16 | 25.08 |
| 0124_03 | 28.50 | 51.36 | 28.70 | 44.55 | 28.95 | 58.21 | 29.06 | 44.14 | 29.31 | 61.58 | 29.56 | 46.19 | 30.55 | 38.04 |
| 0813_05 | 21.36 | 108.59 | 28.90 | 38.32 | 29.00 | 53.61 | 29.02 | 40.27 | 29.39 | 57.22 | 29.87 | 41.25 | 30.85 | 33.22 |
| Avg | 26.10 | 54.56 | 26.53 | 42.76 | 27.26 | 56.82 | 27.10 | 42.36 | 27.56 | 58.24 | 27.85 | 42.84 | 28.78 | 35.24 |

generalization, this raises a speed–accuracy trade-off, and striking the optimal balance requires further investigation.

While our approach can still match or even surpass baselines under extreme pose conditions, particularly with loose-fitting clothing and long video sequences, it may significantly distort the human body and introduce visually distracting artifacts as shown in Fig. L. Future work could address this issue by integrating physical priors into the dynamics module to enforce realism, or by leveraging motion foundation models to better handle complex pose variations.

# F    SOCIAL IMPACTS

Our research has the potential to substantially advance the efficiency and accessibility of human avatar modeling pipelines, helping to address the limitations of supervised datasets where certain demographics, body types, or activities remain underrepresented. By requiring only a single monocular video, our approach lowers the barrier to creating high-quality avatars and broadens the scope of possible applications.

However, this very accessibility also heightens the risk of misuse, including the unauthorized generation of 3D models without an individual's consent. Such practices raise serious ethical and privacy concerns. It is therefore imperative that any deployment of our method strictly follows informed consent protocols, complies with legal and institutional regulations, and incorporates safeguards against abuse. Beyond technical innovation, responsible use must remain a central principle to ensure that advances in avatar modeling benefit individuals and society without compromising their rights or autonomy.

# G    THE USE OF LARGE LANGUAGE MODELS

As recommended by the ICLR committee, we clarify that Large Language Models (LLMs) were used solely for text polishing in this paper, and not for research ideation or initial manuscript writing.

Table I: **Novel-view synthesis comparisons on DNA-Rendering** with FID and KID scores. Cell color indicates best.

| | Dyco | | LS-Avatar | | GauHuman | | GoMAvatar | | ToMie | | 3DGS-Avatar | | **Ours** | |
|---|---|---|---|---|---|---|---|---|---|---|---|---|---|---|
| | FID↓ | KID↓ | FID↓ | KID↓ | FID↓ | KID↓ | FID↓ | KID↓ | FID↓ | KID↓ | FID↓ | KID↓ | FID↓ | KID↓ |
| 0007_04 | 134.99 | 51.87 | 113.90 | 34.78 | 187.02 | 119.88 | 130.05 | 52.97 | 183.36 | 113.08 | 120.91 | 41.55 | 99.28 | 21.85 |
| 0010_03 | 75.55 | 11.95 | 60.44 | 4.39 | 104.84 | 51.85 | 89.80 | 32.08 | 108.35 | 53.46 | 69.01 | 11.70 | 54.42 | 1.71 |
| 0017_11 | 63.16 | 10.59 | 43.66 | 4.29 | 96.78 | 43.45 | 60.38 | 9.18 | 106.16 | 58.06 | 65.68 | 15.07 | 46.76 | 3.50 |
| 0025_09 | 41.70 | 3.34 | 35.42 | 4.12 | 92.77 | 64.54 | 45.57 | 8.64 | 86.70 | 57.30 | 54.97 | 16.18 | 35.05 | 3.58 |
| 0042_12 | 74.98 | 48.43 | 69.99 | 45.70 | 135.29 | 134.72 | 84.51 | 66.39 | 141.96 | 140.03 | 126.86 | 107.61 | 47.79 | 22.70 |
| 0044_10 | 38.99 | 6.69 | 46.23 | 8.36 | 83.30 | 55.56 | 51.49 | 18.86 | 83.78 | 52.98 | 44.34 | 14.83 | 35.77 | 7.80 |
| 0124_03 | 81.80 | 26.80 | 62.14 | 12.96 | 130.92 | 98.08 | 81.92 | 33.67 | 146.84 | 111.71 | 91.89 | 37.55 | 56.90 | 11.31 |
| 0813_05 | 245.10 | 259.89 | 79.11 | 27.21 | 179.03 | 137.18 | 87.77 | 37.19 | 166.05 | 134.45 | 114.32 | 60.36 | 72.73 | 24.89 |
| Avg | 94.53 | 52.45 | 63.74 | 16.94 | 126.24 | 88.16 | 78.94 | 32.37 | 127.90 | 90.13 | 86.00 | 38.11 | 56.09 | 12.17 |

Table J: **Novel-pose animation comparisons on DNA-Rendering** with PSNR and LPIPS scores. Cell color indicates best.

| | Dyco | | LS-Avatar | | GauHuman | | GoMAvatar | | ToMie | | 3DGS-Avatar | | **Ours** | |
|---|---|---|---|---|---|---|---|---|---|---|---|---|---|---|
| | PSNR↑ | LPIPS↓ | PSNR↑ | LPIPS↓ | PSNR↑ | LPIPS↓ | PSNR↑ | LPIPS↓ | PSNR↑ | LPIPS↓ | PSNR↑ | LPIPS↓ | PSNR↑ | LPIPS↓ |
| 0007_04 | 21.79 | 106.87 | 22.30 | 98.33 | 22.78 | 103.98 | 20.52 | 107.04 | 22.25 | 107.04 | 22.40 | 90.88 | 22.64 | 77.41 |
| 0010_03 | 21.91 | 85.28 | 21.18 | 72.59 | 21.78 | 82.57 | 21.57 | 72.79 | 21.82 | 83.50 | 21.67 | 75.16 | 21.80 | 66.09 |
| 0017_11 | 23.94 | 62.82 | 23.74 | 48.02 | 24.29 | 57.38 | 23.78 | 49.57 | 24.38 | 61.97 | 23.91 | 51.54 | 24.24 | 46.02 |
| 0025_09 | 25.99 | 57.72 | 26.40 | 49.65 | 26.95 | 67.58 | 26.07 | 51.77 | 27.55 | 63.51 | 27.09 | 50.64 | 27.88 | 42.43 |
| 0042_12 | 25.50 | 40.99 | 24.43 | 41.40 | 25.45 | 47.40 | 24.87 | 40.77 | 25.60 | 49.84 | 26.04 | 43.84 | 26.89 | 28.62 |
| 0044_10 | 25.29 | 52.41 | 25.60 | 46.26 | 26.04 | 54.66 | 25.82 | 46.69 | 25.72 | 55.09 | 25.38 | 44.85 | 25.05 | 47.07 |
| 0124_03 | 24.92 | 89.35 | 25.72 | 77.86 | 25.69 | 89.86 | 25.39 | 78.15 | 25.67 | 93.91 | 25.67 | 81.06 | 25.80 | 75.42 |
| 0813_05 | 20.54 | 137.71 | 25.52 | 64.37 | 25.72 | 77.49 | 25.28 | 70.34 | 25.64 | 81.61 | 25.36 | 69.73 | 25.75 | 63.37 |
| Avg | 23.74 | 79.14 | 24.36 | 59.79 | 24.84 | 71.91 | 24.16 | 64.26 | 24.83 | 74.56 | 24.69 | 63.46 | 25.01 | 55.80 |

Table K: **Novel-pose animation comparisons on DNA-Rendering** with FID and KID scores. Cell color indicates best.

| | Dyco | | LS-Avatar | | GauHuman | | GoMAvatar | | ToMie | | 3DGS-Avatar | | **Ours** | |
|---|---|---|---|---|---|---|---|---|---|---|---|---|---|---|
| | FID↓ | KID↓ | FID↓ | LPIPS↓ | FID↓ | LPIPS↓ | FID↓ | LPIPS↓ | FID↓ | LPIPS↓ | FID↓ | LPIPS↓ | FID↓ | LPIPS↓ |
| 0007_04 | 313.50 | 219.24 | 269.93 | 171.97 | 284.08 | 202.93 | 246.32 | 144.87 | 281.16 | 181.57 | 260.01 | 128.95 | 212.96 | 88.51 |
| 0010_03 | 222.42 | 165.65 | 160.12 | 95.01 | 197.22 | 149.98 | 173.73 | 109.10 | 198.35 | 153.08 | 205.93 | 136.15 | 165.57 | 97.68 |
| 0017_11 | 114.82 | 64.87 | 67.28 | 28.21 | 118.64 | 79.00 | 69.09 | 21.82 | 124.81 | 87.50 | 90.19 | 41.60 | 68.57 | 26.68 |
| 0025_09 | 56.78 | 11.58 | 53.78 | 13.75 | 108.34 | 78.27 | 61.35 | 16.05 | 104.65 | 69.22 | 70.94 | 26.60 | 52.77 | 12.19 |
| 0042_12 | 74.79 | 51.35 | 61.99 | 37.26 | 125.93 | 130.78 | 81.46 | 62.33 | 141.49 | 143.34 | 120.29 | 108.71 | 47.27 | 22.53 |
| 0044_10 | 126.99 | 84.99 | 126.31 | 84.29 | 147.86 | 131.07 | 126.77 | 77.61 | 174.92 | 165.94 | 121.88 | 72.64 | 116.43 | 58.64 |
| 0124_03 | 135.11 | 94.78 | 136.13 | 113.08 | 166.85 | 151.78 | 117.00 | 84.12 | 184.78 | 189.45 | 114.61 | 66.81 | 102.61 | 57.83 |
| 0813_05 | 246.05 | 273.77 | 89.21 | 36.07 | 198.72 | 169.88 | 113.49 | 70.61 | 205.13 | 181.05 | 151.96 | 95.10 | 98.41 | 34.57 |
| Avg | 161.31 | 120.78 | 120.22 | 72.46 | 168.45 | 136.71 | 123.65 | 73.31 | 176.91 | 146.40 | 141.98 | 84.57 | 108.07 | 49.83 |

Table L: **Novel-view synthesis comparisons on MVHumanNet** with the mean angular error and accuracy within $22.5°$. Cell color indicates best.

| | Dyco | | LS-Avatar | | GauHuman | | GoMAvatar | | ToMie | | 3DGS-Avatar | | **Ours** | |
|---|---|---|---|---|---|---|---|---|---|---|---|---|---|---|
| | Mean↓ | 22.5°↑ | Mean↓ | 22.5°↑ | Mean↓ | 22.5°↑ | Mean↓ | 22.5°↑ | Mean↓ | 22.5°↑ | Mean↓ | 22.5°↑ | Mean↓ | 22.5°↑ |
| 100846 | 73.17 | 5.09 | 38.70 | 29.44 | 63.69 | 12.03 | 54.40 | 6.77 | 71.42 | 7.19 | 69.06 | 7.10 | 24.33 | 62.87 |
| 100990 | 50.04 | 19.36 | 36.64 | 33.69 | 56.84 | 16.28 | 55.54 | 4.53 | 59.17 | 13.29 | 61.67 | 11.60 | 22.76 | 66.38 |
| 102107 | 47.36 | 22.48 | 30.03 | 47.30 | 54.00 | 21.13 | 56.43 | 4.35 | 60.23 | 14.47 | 58.56 | 13.53 | 20.32 | 72.10 |
| 102145 | 47.39 | 23.05 | 35.13 | 35.00 | 58.32 | 15.64 | 52.45 | 5.68 | 57.98 | 15.03 | 59.58 | 12.83 | 19.54 | 73.63 |
| 103708 | 52.14 | 18.23 | 39.48 | 31.15 | 55.61 | 17.39 | 53.77 | 4.79 | 60.19 | 14.64 | 60.83 | 12.40 | 26.31 | 57.31 |
| 200173 | 52.48 | 17.09 | 33.56 | 39.96 | 61.63 | 14.48 | 60.73 | 2.53 | 62.86 | 12.20 | 57.16 | 15.05 | 25.12 | 59.34 |
| 204112 | 42.88 | 31.26 | 29.26 | 47.66 | 51.71 | 22.92 | 59.94 | 3.09 | 51.57 | 22.69 | 52.22 | 20.87 | 19.49 | 72.92 |
| 204129 | 51.65 | 17.38 | 35.60 | 33.40 | 58.37 | 14.30 | 60.78 | 3.30 | 60.79 | 11.97 | 55.63 | 14.89 | 19.74 | 71.08 |
| Avg | 52.14 | 19.24 | 34.80 | 37.20 | 57.52 | 16.77 | 56.76 | 4.38 | 60.53 | 13.94 | 59.34 | 13.53 | 22.20 | 66.95 |

Table M: **Novel-view synthesis comparisons on DNA-Rendering** with the mean angular error and accuracy within 22.5°. Cell color indicates best .

| | Dyco | | LS-Avatar | | GauHuman | | GoMAvatar | | ToMie | | 3DGS-Avatar | | **Ours** | |
|---|---|---|---|---|---|---|---|---|---|---|---|---|---|---|---|
| | Mean ↓ | 22.5° ↑ | Mean ↓ | 22.5° ↑ | Mean ↓ | 22.5° ↑ | Mean ↓ | 22.5° ↑ | Mean ↓ | 22.5° ↑ | Mean ↓ | 22.5° ↑ | Mean ↓ | 22.5° ↑ |
| 0007_04 | 59.95 | 11.33 | 46.62 | 24.82 | 66.72 | 12.80 | 52.59 | 6.40 | 62.20 | 15.91 | 60.22 | 12.32 | 24.39 | 63.05 |
| 0010_03 | 48.82 | 24.99 | 43.44 | 34.81 | 58.32 | 17.84 | 54.24 | 4.80 | 57.70 | 17.82 | 55.75 | 18.02 | 23.92 | 61.97 |
| 0017_11 | 53.11 | 20.20 | 45.96 | 31.03 | 65.20 | 14.42 | 59.39 | 4.99 | 66.96 | 13.23 | 54.65 | 19.87 | 20.13 | 75.32 |
| 0025_09 | 49.52 | 24.06 | 39.58 | 40.94 | 63.61 | 15.00 | 56.60 | 5.57 | 61.47 | 16.07 | 59.09 | 13.79 | 18.91 | 76.28 |
| 0042_12 | 54.49 | 18.54 | 47.36 | 25.53 | 68.16 | 9.12 | 53.13 | 7.92 | 70.02 | 8.42 | 62.29 | 12.34 | 17.97 | 79.31 |
| 0044_10 | 39.87 | 36.70 | 36.84 | 42.96 | 62.77 | 15.00 | 53.64 | 5.17 | 58.99 | 18.21 | 57.58 | 17.31 | 17.18 | 79.06 |
| 0124_03 | 54.53 | 18.19 | 44.77 | 32.77 | 72.04 | 9.12 | 56.00 | 5.33 | 71.54 | 8.85 | 58.84 | 14.93 | 20.68 | 72.04 |
| 0813_05 | 66.24 | 9.36 | 44.50 | 34.95 | 74.06 | 8.91 | 56.98 | 6.30 | 73.13 | 9.61 | 60.76 | 14.98 | 20.51 | 74.23 |
| Avg | 53.32 | 20.42 | 43.63 | 33.48 | 66.36 | 12.78 | 55.32 | 5.81 | 65.25 | 13.52 | 58.65 | 15.45 | 20.46 | 72.66 |

GT  DyCo  LS-Avatar  GoMAvatar  GauHuman  ToMiE  3DGS-Avatar  Ours

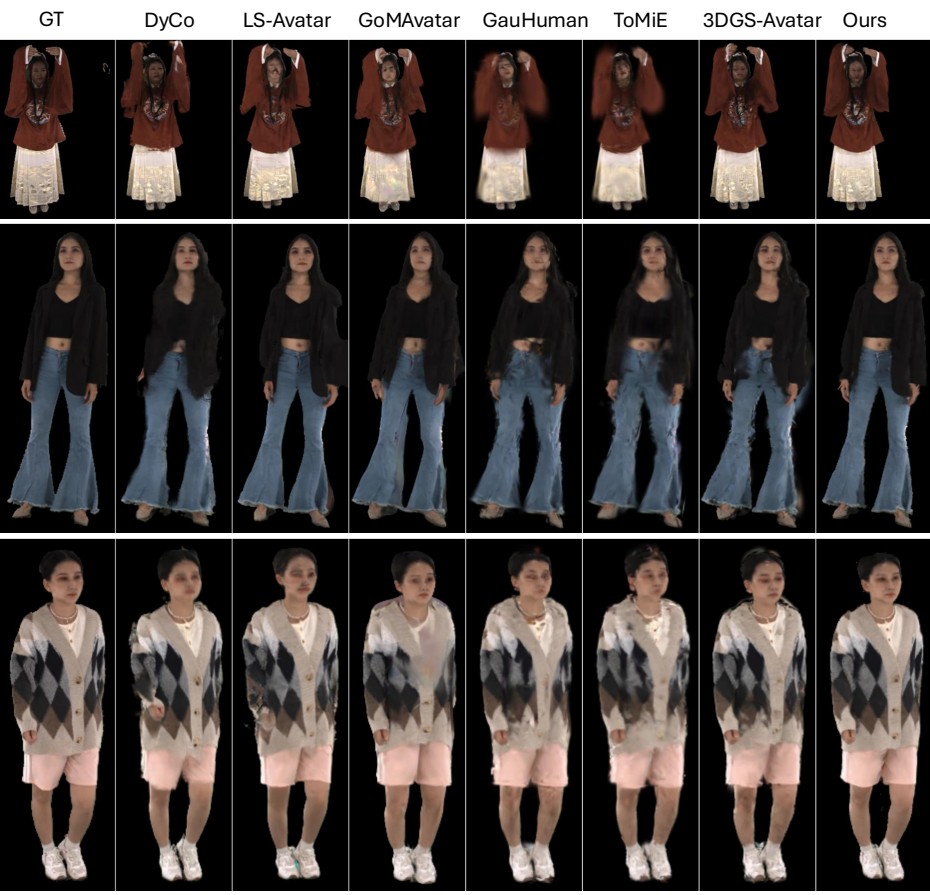

Figure D: Novel View Synthesis on the DNA-Rendering dataset.

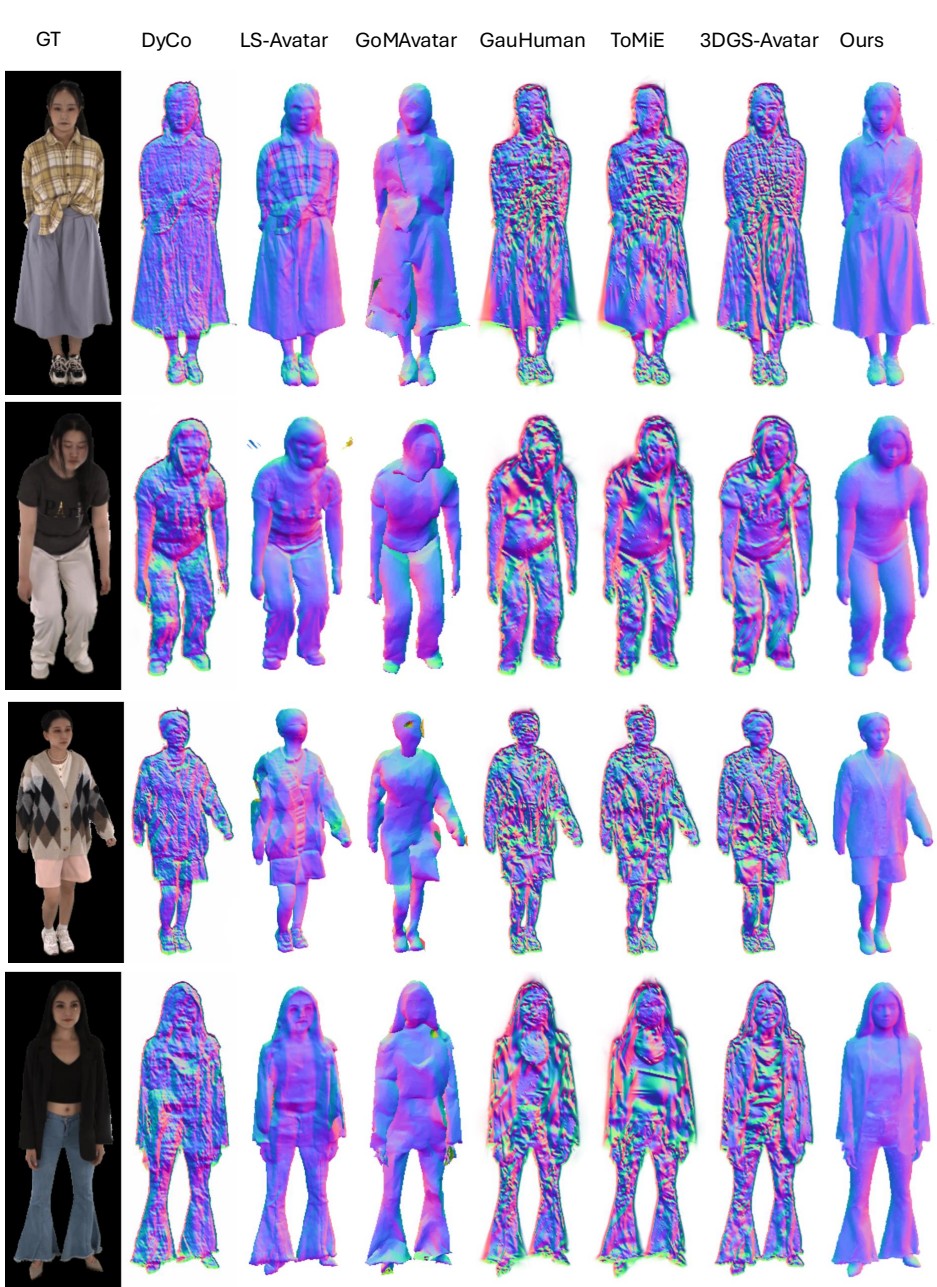

Figure E: **Normal Estimation on DNA-Rendering.** Compared to existing baselines, our method can more successfully reconstruct smooth large-scale structures and fine-grained patterns in all cases.

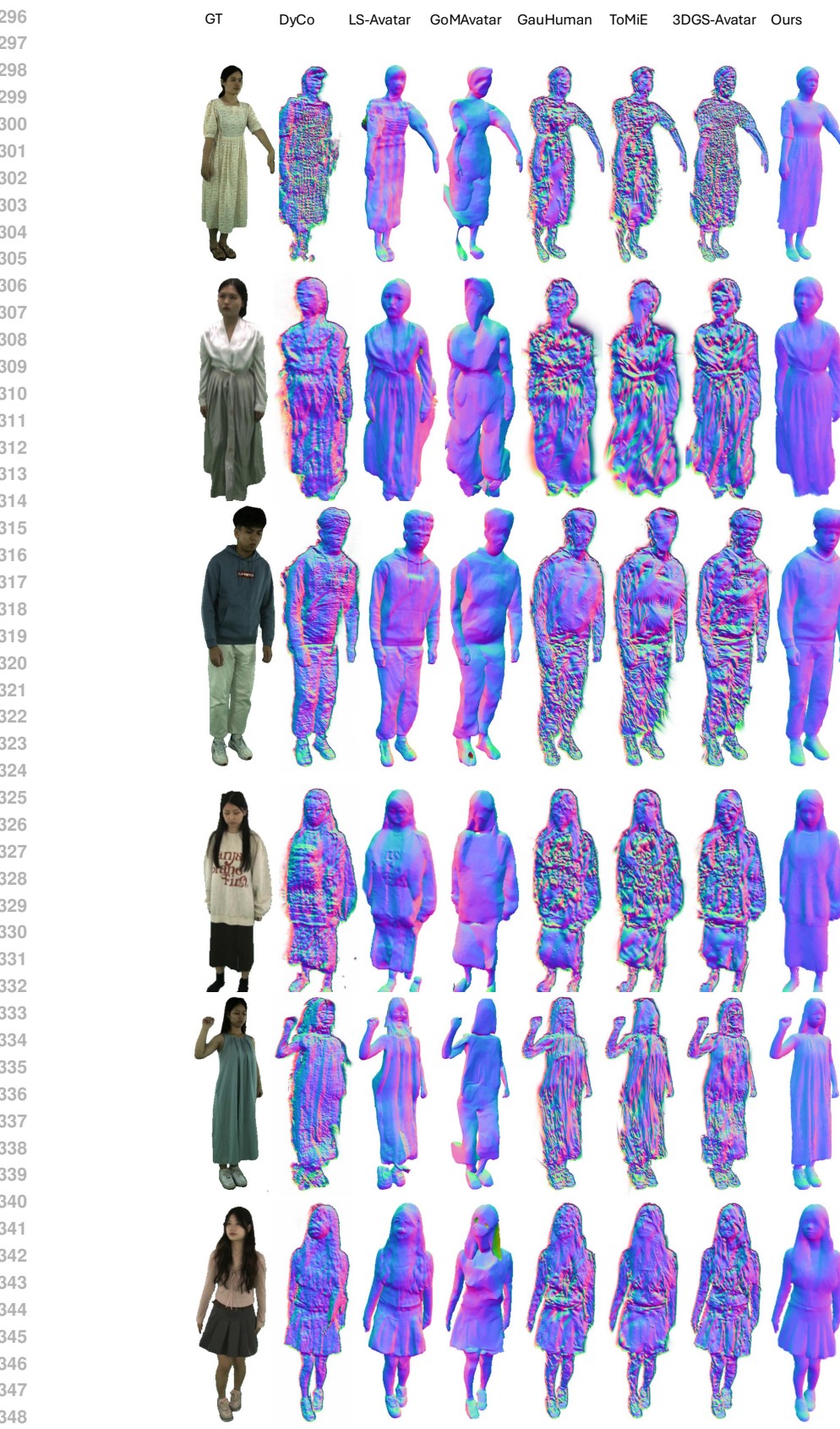

Figure F: **Normal Estimation on MVHumanNet.** Compared to existing baselines, our method can more successfully reconstruct smooth large-scale structures and fine-grained patterns in all cases.

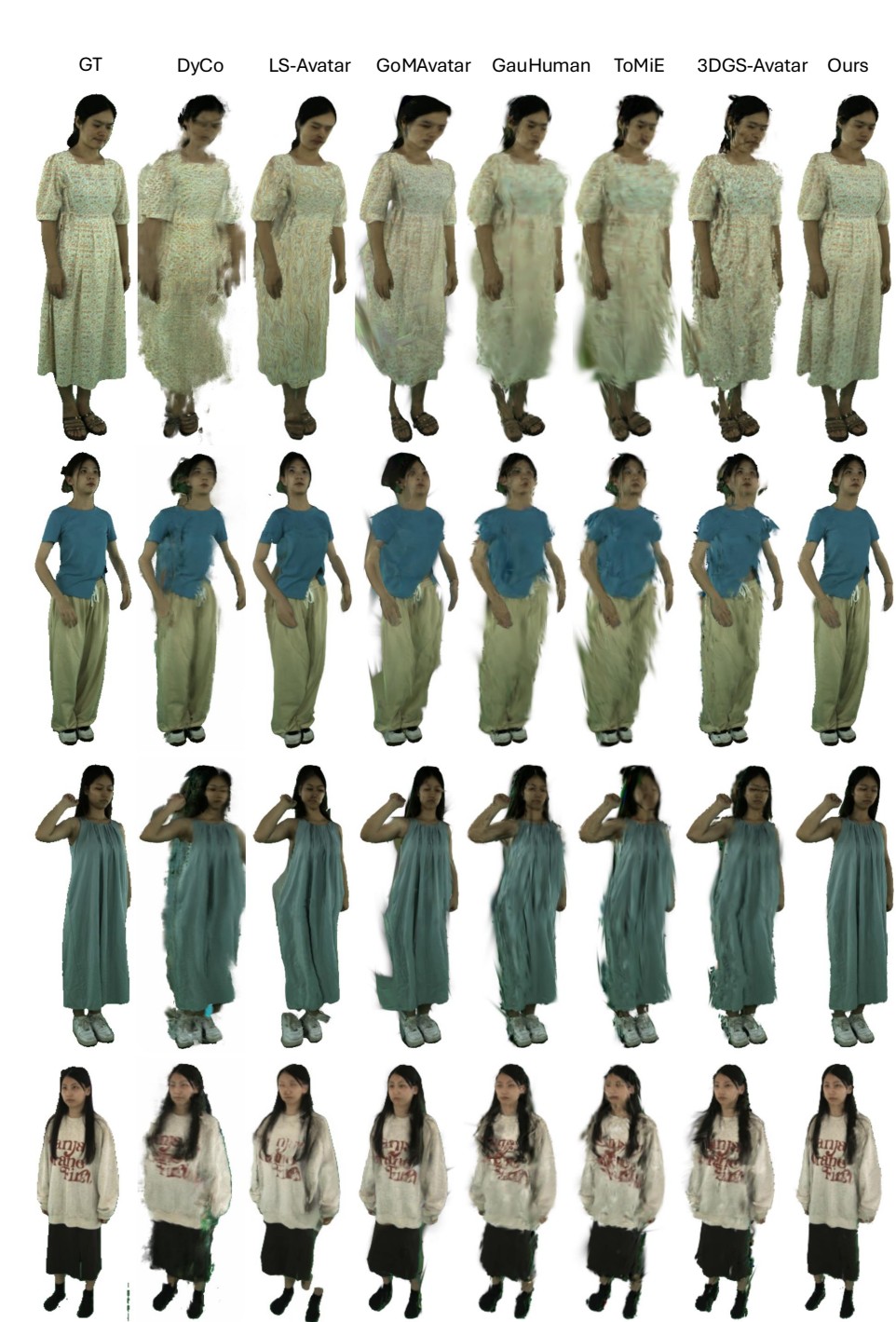

Figure G: Novel View Synthesis on the MVHumanNet dataset.

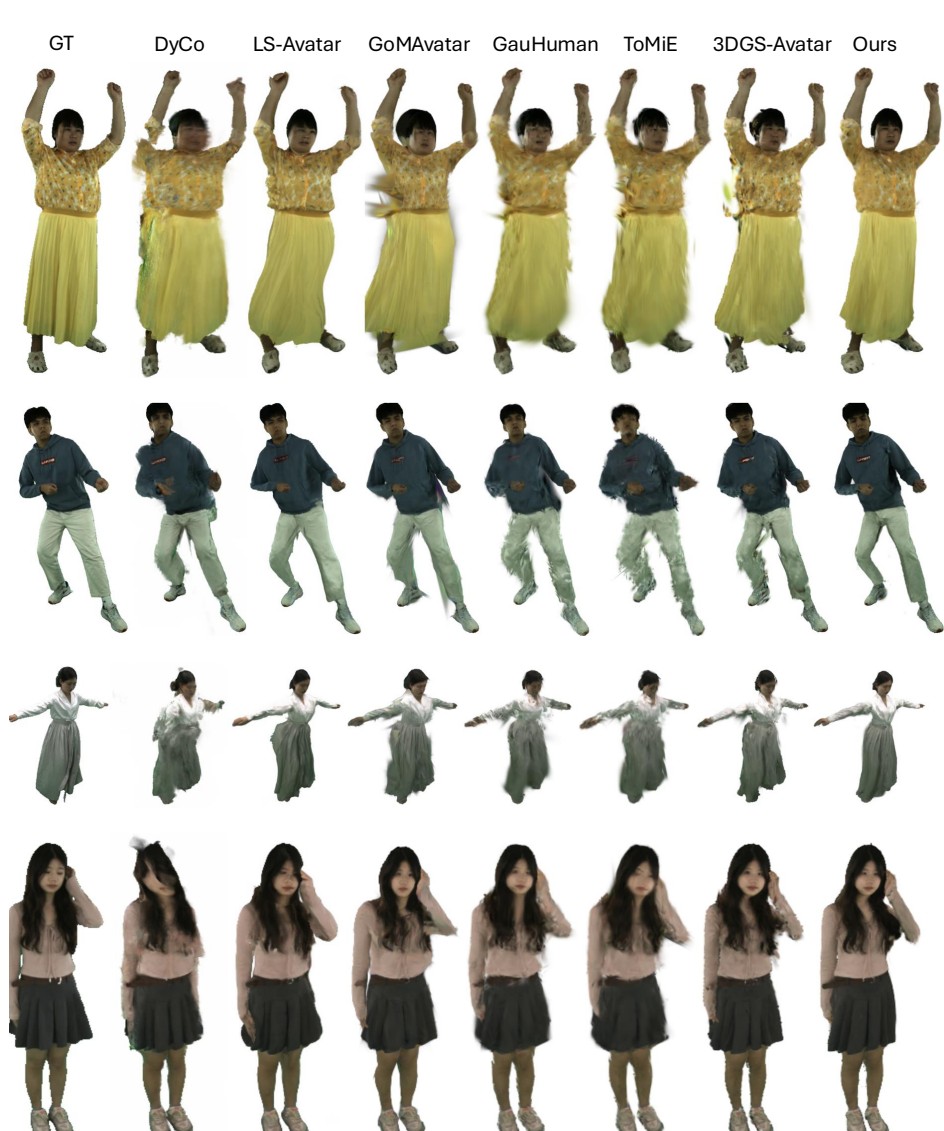

Figure H: Novel Pose Animation on the MVHumanNet dataset.

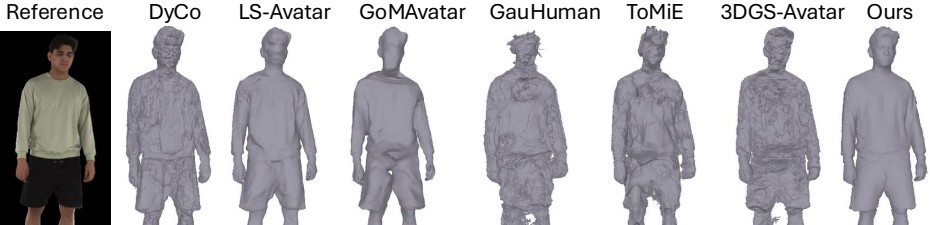

Figure I: Mesh generation on the DNA-Rendering dataset.

GT    DyCo    LS-Avatar    GoMAvatar    GauHuman    ToMiE    3DGS-Avatar    Ours

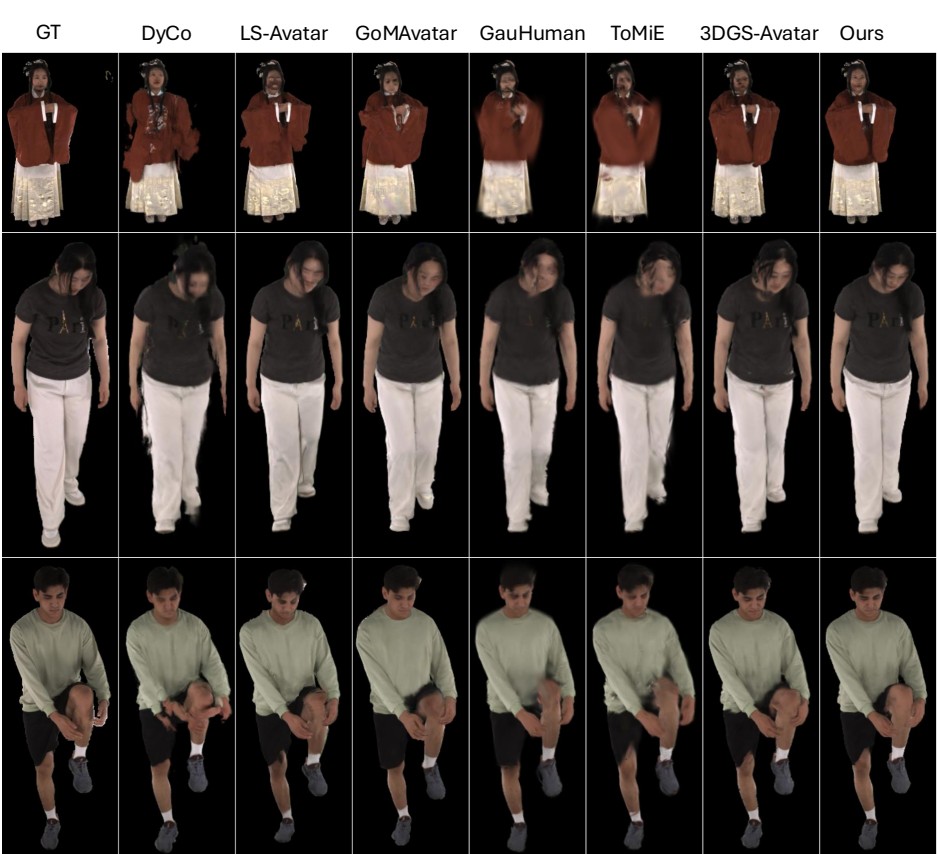

Figure J: Novel Pose Animation on the DNA-Rendering dataset.

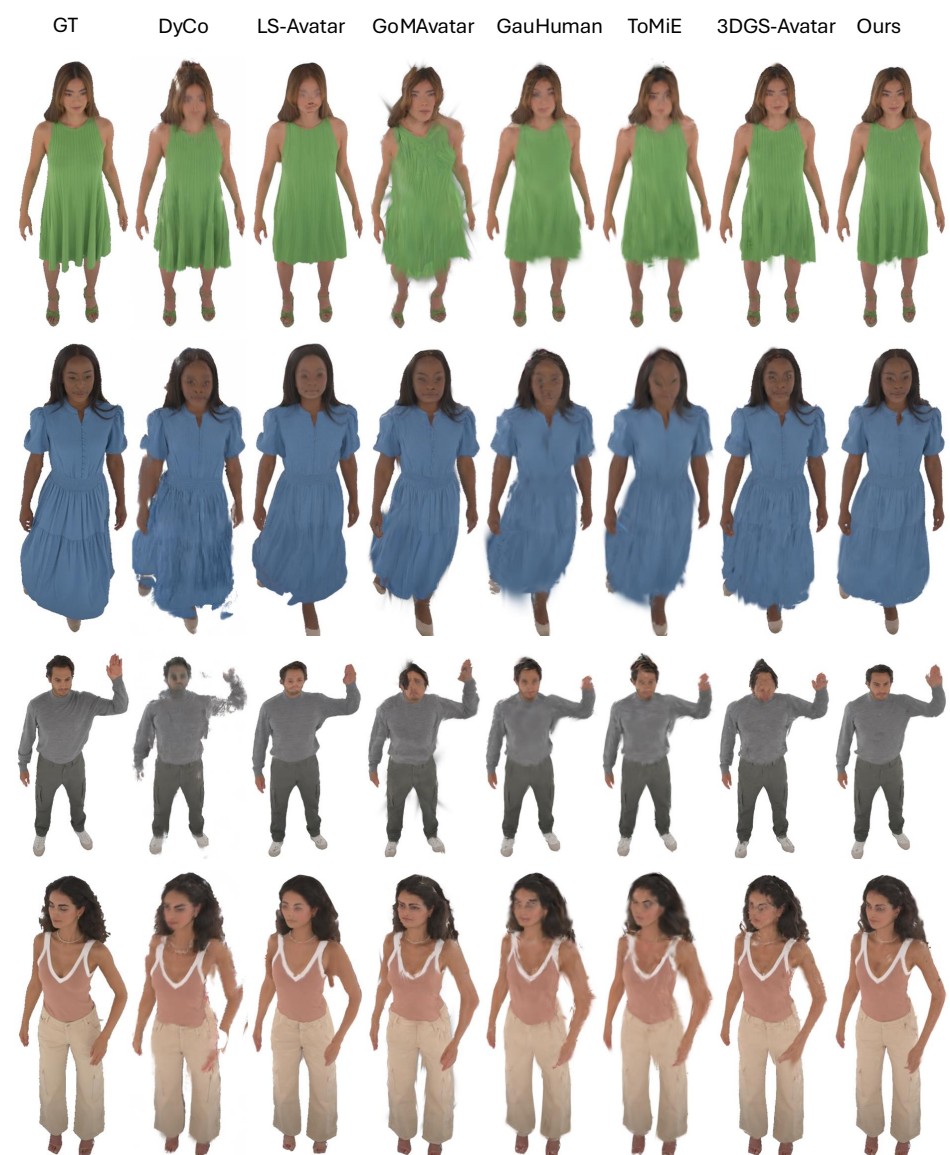

Figure K: Image rendering comparisons on the ActorsHQ dataset. The top two rows show novel view synthesis results, while the bottom row illustrates novel pose animation.

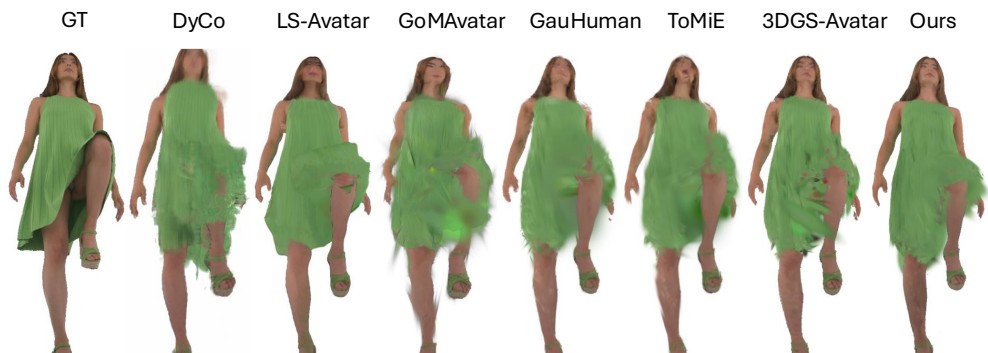

Figure L: **Failure cases on loose-fitting cloth.** While our method better preserves the realistic fine-grained patterns such as human face and hands, none of all methods can faithfully model the loose clothing in challenging poses from monocular videos.

