# OpenReview forum: "Motion-Aware Surface Smoothing for Monocular Avatar Representations"
_ICLR.cc/2026/Conference — Submitted to ICLR 2026_

### Official Review · Reviewer_3cMT · 2025-10-17

**Soundness:** 3
**Presentation:** 3
**Contribution:** 2
**Rating:** 4
**Confidence:** 4

**Summary:**

The focus of the paper is on human modeling from monocular videos, especially from sparse frames.

The main contribution is a regularization term to encourage the smoothness of the surface from a camera view. Since the regularization is computed in 2D space after rendering the avatar articulated by a certain pose, the smoothness is disrupted at occlusion boundaries and in regions with holes. Therefore it further weights the regularization based on a distance between the corresponding points in the canonical space.

It furthers adds a smoothness-based heuristic in adaptive density control: If a Gaussian contributes more to smooth surface, it will be more likely to be densified.

It experiments on MVHumanNet, DNA-Rendering, ActorsHQ and Youtube videos. It shows superior results in sparse-view settings compared to the baselines.

**Strengths:**

* The paper tackles the important problem of human modeling from sparse inputs. While dense inputs may require cumbersome data collection, sparse views as inputs are more practical.
* The paper is clearly written and easy to follow. The methods are well motivated, e.g., why to weight the difference of depths based on the distance of Gaussians in canonical space.
* The idea of regularizing the surface smoothness is interesting. It identifies the cases (occlusions and holes) where smoothness regularization does not hold and proposes the fix for such scenarios. It is novel and reasonable to considering the contribution of a Gaussian to the smooth surface in adaptive density control.
* It experiments on various dataset and challenging in-the-wild videos, showing that the method is widely applicable.

**Weaknesses:**

* The paper does not compare to iHuman, which also tackles the problem of human modeling from sparse inputs. iHuman [1] can model with as few as 6 views. Therefore, it is an important baseline.

[1] iHuman: Instant Animatable Digital Humans From Monocular Videos, Paudel et al., ECCV 2024.

* While the contributions lie in the smooth regularization and geometry-aware adaptive density control, the biggest improvement comes from replacing 3DGS with GoF and the normal map supervision from Sapiens.

* When applying the smoothness regularization, the authors choose the observation space where occlusions disrupt the smoothness assumption. However, such difficulties may not exist in the canonical space, where the avatar is carefully poses to avoid occlusions as much as possible. It is not clear to me the benefits of using observation space.

* The image sizes in the supplementary videos vary continuously, making it hard to identify the potential flickering and multiview consistency.

* One of the key challenges of modeling from monocular videos is the incomplete observation. The monocular videos may not depict every aspect of the avatar. The paper does not show multiview rendering, therefore it's hard to tell how it hallucinates in unseen regions.

**Questions:**

My main concerns are the missing baseline and limited improvement from the contributions, as listed in the weakness. Besides, the followings are minor questions and suggestions:

* When locating the nearest Gaussian along the ray, does the method also consider the value of the opacity? For example, if a Gaussian very close to the camera is almost transparent, will it be used as depth? If so, the depth may be misleading.

* I suggest to be careful with the word "sparse inputs" as sometimes it refers to even sparser input like 3 images. The settings of the paper are monocular videos which use over 100 frames for training.

Potential typos:
* In Eq. 3, the right most $x_o$ should be $x_c$.
* In Eq. 7, is it supposed to be $S_i=\sum_{j}w_j \|I_j^d - I_i^d\|$ rather than $S_i=\sum_{j}w_j (I_j^d - I_i^d)$?

---

> ### Author Response · Authors · 2025-11-26
> **Response to Reviewer 3cMT**
>
> We thank the reviewer for carefully evaluating our paper and providing valuable feedback. We address the concerns in both the common responses and the section below, and we remain open to any further questions or discussion.
>
> **Q1: The benefits of using observation space to apply the smoothness regularization.**
>
> We emphasize that our goal is to render visually reasonable images and synthesize smooth and compact 3D surfaces for each input frame. Computing the smoothness regularization in the canonical space cannot regularize the learning of rigid deformation and non-rigid deformation explicitly and thus might yield inferior generalization to monocular inputs. To demonstrate the effectiveness of regularizing in observation space, we perform the surface smoothing based on the canonical space for comparisons. Specifically, we remain everything in our framework unchanged except that we remove the weight $w_j$ and compute the smoothness map $S$ for canonical Gaussians. The quantitative metrics are provided below and our full model consistently outperforms the ablated model named as **Canonical Smoothing** across all metrics.
>
> |            | PSNR  | SSIM   | LPIPS  |
> |:----------:|:-----:|:------:|:------:|
> | Novel View |       |        |        |
> | Canonical Smoothing | 25.16 | 0.966 | 34.48  |
> | Full Model | **25.30** | **0.968** | **32.10**  |
> | Novel Pose |       |        |        |
> | Canonical Smoothing | 22.98 | 0.956 | 46.58 |
> | Full Model | **23.00** | **0.957** | **44.56**  |
>
> We additionally provide the qualitative comparisons in Figure B for a comprehensive demonstration. We can see that regularizing surfaces in the canonical space is prone to generate floating artifacts in empty space and distort the shape boundaries while our full model can preserve more fluent geometric contours.
>
> **Q2: The image sizes in the supplementary videos vary continuously.**
>
> Thank you for pointing this out. We have revised the format of our supplementary video to make it visualize more stably.
>
> **Q3: It's hard to tell how it hallucinates in unseen regions.**
>
> Thank you for your comments. We have added a new section named as **360$\degree{}$ View Visualization** in the supplementary video to exemplify the generation of the whole human character. It is clear that our method can successfully generalize to unseen regions under the multi-view rendering setting.
>
> **Q4: Does the method also consider the value of the opacity when locating the nearest Gaussian along the ray?**
>
> Yes, we do consider the opacity value when computing depth maps. In short, we set a threshold for the opacity of each Gaussian and discard those Gaussians with very low opacity values to compute more accurate depth maps.
>
>
> **Q5: I suggest to be careful with the word "sparse inputs".**
>
> Thank you for your suggestions. We have replaced the word "sparse inputs" with "monocular videos" in our revised paper.
>
>
> **Q6: Potential typos.**
>
> Thank you for pointing these out. We have addressed them in the revised paper.

---

### Official Review · Reviewer_qjx3 · 2025-10-25

**Soundness:** 3
**Presentation:** 2
**Contribution:** 3
**Rating:** 4
**Confidence:** 4

**Summary:**

The paper addresses the task of monocular video-based human reconstruction by introducing a motion-aware surface smoothing framework. The key contribution lies in modulating the Adaptive Density Control (ADC) and supervising Gaussian motions using unseen virtual camera viewpoints. By explicitly regularizing surface smoothness through depth-based constraints, the method enhances the geometric consistency of 3D Gaussian Splatting. Evaluations across five public datasets demonstrate improved performance over recent state-of-the-art approaches in novel view synthesis, pose animation, and 3D shape reconstruction.

**Strengths:**

- The motion-aware surface smoothing mechanism effectively enforces spatial regularization of Gaussians using rendered depth maps. Combined with the geometry-aware Adaptive Density Control, it results in smoother and more compact surface geometry.
- The proposed method not only improves visual fidelity but also produces plausible mesh reconstructions, outperforming previous Gaussian-based avatar models in terms of geometric coherence and normal consistency.

**Weaknesses:**

- While the paper claims improvements in mesh reconstruction quality, no mesh-level evaluation metrics such as *Chamfer Distance* or *Point-to-Surface (P2S)* error are reported. Including these would quantitatively support the claim that the method improves geometric accuracy. A comparison against recent mesh-based human reconstruction works would further strengthen this aspect.

- In ActorsHQ novel pose evaluation, the performance drops compared to several baselines. The authors briefly acknowledge this in the appendix but do not provide an explicit analysis. An explanation regarding the causes would be valuable.

- The supplementary video exhibits minor stretching artifacts and floating points around the legs (notably at 1m 5–8s). Since the framework is designed to promote spatial smoothness and motion consistency, a discussion on why these artifacts occur would clarify the limitations.

- The comparison set is limited. Several recent monocular video-based reconstruction works with released code, such as *HUGS* (Kocabas et al., 2024), and *Expressive Gaussian Avatar* (Moon et al., 2024), should be included for a fairer comparison.

- Missing relevant recent references related to mesh-based / gaussian-based reconstruction:
    - [1] Moon, Gyeongsik, et al. *“Expressive Whole-body 3D Gaussian Avatar.”* **ECCV 2024.**

    - [2] Shin, Jisu, et al. *“CanonicalFusion: Generating Drivable 3D Human Avatars from Multiple Images.”* **ECCV 2024.**

    - [3] Shao, Zhijing, et al. *“SplattingAvatar: Realistic Real-time Human Avatars with Mesh-embedded Gaussian Splatting.”* **CVPR 2024.**

    - [4] Svitov, David, et al. *“HAHA: Highly Articulated Gaussian Human Avatars with Textured Mesh Prior.”* **ACCV 2024.**

**Questions:**

- Could the authors provide training time comparisons with other baselines in the main paper or appendix? Since 3DGS-based methods often emphasize efficiency, this comparison would contextualize the proposed improvements.

- What are the common failure cases (e.g., loose garments, extreme poses, occlusions, or inaccurate SMPL initialization)? Including a small visualization of typical failure cases would help readers understand the framework’s limitation.

---

> ### Author Response · Authors · 2025-11-26
> **Response to Reviewer qjx3**
>
> We thank the reviewer for carefully evaluating our paper and providing valuable feedback. We address the concerns in both the common responses and the section below, and we remain open to any further questions or discussion.
>
> **Q1: No mesh-level evaluation metrics are reported.**
>
> Thank you for your suggestions. As suggested, we follow the settings of our latest baseline LS-Avatar (ICLR 2025) and report the quantitative comparisons with **Chamfer Distance** and **Normal Consistency** on the MVHumanNet sequences.
>
> As reported in the table below, our method achieves better quantitative results than all baselines in terms of both metrics which shows our superior geometry reconstruction capability.
>
> |                    |  DyCo | LS-Avatar | GauHuman | GoM-Avatar | ToMie | 3DGS-Avatar |    Ours   |
> |:------------------:|:-----:|:---------:|:--------:|:----------:|:-----:|:-----------:|:---------:|
> |  Chamfer Distance  |  6.02 |    5.18   |   14.17  |    7.30    | 11.71 |    12.00    |  **4.14** |
> | Normal Consistency | 0.707 |   0.786   |   0.663  |    0.768   | 0.678 |    0.652    | **0.803** |
>
> Please note that, GoM-Avatar (CVPR 2024) and LS-Avatar (ICLR 2025) are two geometry-focused baselines which also aim to reconstruct realistic human surfaces. Based on the additional results above, together with the findings reported in both the main paper and the response, our method consistently outperforms existing baselines across novel view synthesis, novel pose animation and surface reconstruction.
>
> **Q2: In ActorsHQ novel pose evaluation, the performance drops compared to several baselines.**
>
> Compared with other datasets used in our experiments, such as MVHumanNet and DNA-Rendering, the ActorsHQ sequences have higher image resolution and contain much finer visual details. In addition, ActorsHQ includes more challenging human poses that are difficult to train, especially when combined with loosely fitted clothing.
>
> For novel pose animation, the test poses can differ significantly from those seen during training. Thus the model must learn not only human motion but also complex non-rigid cloth deformations, which may require explicit physics-based modeling to generalize well. As discussed in the Limitation section, incorporating such physics priors into our smoothness regularization is beyond the scope of this submission and is an interesting direction for future work. Nevertheless, even under these challenging conditions, our method still achieves performance that is on par with or surpasses all baselines, demonstrating strong generalization across diverse testing scenarios.
>
> **Q3: The supplementary video exhibits minor stretching artifacts and floating points around the legs.**
>
> Thank you for pointing this out. Although our method shows strong capability in handling loose clothing through the non-rigid deformation module, its generalization is still limited by the lack of explicit cloth modeling; for example, as seen at 01:45 in the video. Even with surface smoothing to produce better geometric consistency, it remains challenging to fully regularize the highly non-rigid motion of loose garments, which can still lead to artifacts. One possible solution is to adopt a layered representation that models clothing separately from the human body, rather than treating them jointly. However, since the focus of this work is to highlight the effectiveness of the proposed surface smoothing strategy, we leave explicit cloth modeling to future work.
>
> It is worth noting that, even though our method models the body and clothing as a single entity, it consistently outperforms all baselines including ToMie, which explicitly targets loose-clothing scenarios. We will integrate this point into the failure case discussion.
>
> **Q4: Missing some relevant recent references.**
>
> Thank you for raising this question. We have added the four papers you mentioned into the discussions in Section 2.
>
> **Q5: Visualizing typical failure cases.**
>
> Thank you for your comments. We have followed your suggestions and added Figure L in the revised paper to provide more contexts of our framework’s limitations. Consistent with the discussions above, neither existing methods nor our network can produce fully plausible garment deformations. However, our approach is better able to preserve local details, such as the human face and hands.

---

### Official Review · Reviewer_Uer2 · 2025-10-31

**Soundness:** 2
**Presentation:** 3
**Contribution:** 3
**Rating:** 6
**Confidence:** 3

**Summary:**

This paper proposes MASSAR (Motion-Aware Surface Smoothing for Avatar Representations), a novel framework designed to enhance 3D Gaussian Splatting (3DGS) for human avatar modeling from monocular videos.
The core contribution lies in a motion-aware smoothness regularization, which promotes geometrically consistent Gaussian distributions through depth-map-based supervision.
The proposed regularization is applied in three complementary ways: (1) directly as a smoothness loss, (2) integrated with a geometry-aware Adaptive Density Control (ADC) strategy, and (3) extended to a self-supervised virtual-view training scheme.
Through this design, MASSAR achieves improved performance in both novel-pose animation and novel-view rendering.

**Strengths:**

1.The motion-aware surface smoothing term is simple yet effective in enforcing geometric regularization, addressing a known limitation of 3D Gaussian Splatting (3DGS) in sparse-view or monocular settings.

2.The proposed smoothness weight computation in canonical space (Eq. 7) is elegant and interpretable, as it considers the geometric relationships of nearby pixels in both canonical and observation spaces.

**Weaknesses:**

1.The main contribution—smoothness-based regularization—while effective, is a incremental improvment. compare to original depth smooth regularization, the improvement is limit.


2.While ablations show clear improvements, it would be more convincing to include quantitative sensitivity analysis for hyperparameters such as $\sigma_{s} $ and
$\sigma_{c}$

3.The runtime and memory usage of MASSAR are not reported compared to baseline.

4.There is no visualization result without $\{w_{i}\}$ and with $\{w_{i}\}$.

**Questions:**

1.Since different scenes may exhibit varying levels of surface smoothness, how sensitive is the method to this hyperparameter? Is  $\sigma_{s}$ fixed across all datasets, or is it adaptively tuned per case?

2.Could the proposed smoothness regularization be integrated into other Gaussian-based frameworks (e.g., 2DGS), given that some of them already include surface regularization terms? Would such integration require any modification, or is it directly compatible?

---

> ### Author Response · Authors · 2025-11-26
> **Response to Reviewer Uer2**
>
> We thank the reviewer for carefully evaluating our paper and providing valuable feedback. We address the concerns in both the common responses and the section below, and we remain open to any further questions or discussion.
>
> **Q1: Compared to original depth smooth regularization, the improvement of our contribution is limited.**
>
> Thank you for your comments. To more clearly illustrate the effect of our weight computation in Eq. (7), we set $w_j = 1$ in the full model to effectively remove the weighting mechanism as $w/o\ \{w_j\}$, and report the resulting metrics below.
>
> |            | PSNR  | SSIM   | LPIPS  |
> |:----------:|:-----:|:------:|:------:|
> | Novel View |       |        |        |
> |  $w/o\ \{w_j\}$ | 24.71 | 0.965 | 34.35  |
> | Full Model | **25.30** | **0.968** | **32.10**  |
> | Novel Pose |       |        |        |
> | $w/o\ \{w_j\}$ | 22.81 | 0.955 | 46.69 |
> | Full Model | **23.00** | **0.957** | **44.56**  |
>
> By simply altering the weight computation strategy, we observe a 0.6 PSNR gain and **roughly a 7% relative improvement in LPIPS** for novel view synthesis. Visually, as shown in Figure B, the model with $w/o\ \{w_j\}$ produces severe distortions along the back contours in both RGB renderings and normal map estimates, whereas the motion-aware surface smoothing produces smoother human boundaries with minimal artifacts. All results highlight the crucial role of the motion-aware weight computation.
>
>
> **Q2: There is no visualization result without $w_i$ and with $w_i$.**
>
> Thank you for your suggestions. We have incorporated the visual comparisons with $w/o\ \{w_j\}$ in Figure B. Being coherent with the discussions in L250-L256, our full model can better preserve the shape contours with limited artifacts.
>
> **Q3: Including quantitative sensitivity analysis for hyperparameters such as $\sigma_s$ and $\sigma_c$.**
>
> Thank you for your suggestions. As recommended, we have evaluated our model with different $\sigma_s$ and $\sigma_c$ pairs, where our full model is denoted as (0.05, 0.02) by setting $\sigma_s = 0.05$ and $\sigma_c = 0.02$. The quantitative metrics are provided below and our model appears to be insensitive to different hyperparameters.
>
> |            | (0.01, 0.02) | (0.03, 0.02) | (0.1, 0.02) | (0.05, 0.1) | (0.05, 0.05) | (0.05, 0.01) | **(0.05, 0.02)** |
> |:----------:|:------------:|:------------:|:-----------:|:-----------:|:------------:|:------------:|:------------:|
> | Novel View |              |              |             |             |              |              |              |
> |    PSNR    |     25.41    |     25.35    |    25.22    |    25.29    |     25.21    |     25.30    |     25.30    |
> |    SSIM    |     0.967    |     0.967    |    0.967    |    0.967    |     0.967    |     0.967    |     0.968    |
> |    LPIPS   |     33.17    |     32.64    |    32.46    |    32.73    |     33.21    |     32.65    |     32.10    |
> | Novel Pose |              |              |             |             |              |              |              |
> |    PSNR    |     23.16    |     23.13    |    23.01    |    23.10    |     23.06    |     23.07    |     23.00    |
> |    SSIM    |     0.956    |     0.956    |    0.957    |    0.957    |     0.956    |     0.956    |     0.957    |
> |    LPIPS   |     45.42    |     45.23    |    44.73    |    44.69    |     45.03    |     44.86    |     44.56    |
>
>
> Additionally, we set the same hyperparameters including $\sigma_s$ and $\sigma_c$  across all datasets to perform the comprehensive comparisons.
>
>
> **Q4: Could the proposed smoothness regularization be integrated into other Gaussian-based frameworks?**
>
> Thank you for raising this interesting question, which we believe warrants further investigation. As discussed in the General Surface Reconstruction section of Section 2, several strategies have been proposed to improve geometry reconstruction in 3D Gaussian Splatting. However, to the best of our knowledge, our work is the first to explicitly address this problem from the perspective of surface smoothing, which is orthogonal to existing approaches.
>
> Given this, we expect that our smoothness regularization could be integrated into other Gaussian-based frameworks without substantial modification. As a self-supervised approach, we anticipate that it may offer greater benefits in sparse-input scenarios (e.g., monocular videos) than in multi-view settings. Nevertheless, how to best leverage our surface smoothing concept for shape reconstruction will require additional future work.

---

### Official Review · Reviewer_KRMV · 2025-11-01

**Soundness:** 2
**Presentation:** 3
**Contribution:** 2
**Rating:** 2
**Confidence:** 4

**Summary:**

This paper tackles the task of avatar reconstruction from a monocular video. The paper follows the strategy of canonical space + deformation (linear blend skinning). To improve the quality, the authors propose to enhance surface smoothness. Specifically, the method encourages 3DGS that are close in the observation space to also be close in the canonical space. Further, it utilizes normal map alignment with priors from foundation models. Experiments on various datasets demonstrate the effectiveness of the proposed approach.

**Strengths:**

- originality-wise: the surface smoothing idea is interesting.
- quality-wise: the final results look promising.
- clarity-wise: the paper is generally well-written.
- significance-wise: reconstructing a human avatar from a monocular video is important for various downstream tasks, e.g., AR/VR.

**Weaknesses:**

1. I am not convinced that the surface smoothing design (Sec. 3.3) contributes most to the model. Actually, when it comes to LPIPS, from Tab. 1, the most important factor is the utilization of Sapiens to compute surface normal (L367).

2. Further, qualitatively, in the ablation (Fig. 5), without the Sapiens' prior, the model produces the lowest quality results, much worse than any other ablations.

3. Eq. (7) encourages close points in the observation space to be close in the canonical space, i.e., $\hat{d}_j = (\tilde{\mu}_j^c - \tilde{\mu}_i^c)$. I am curious whether this is a good signal, as close points in observation space can be really far away in the canonical space. For example, the hand will be close to the feet in the observation space if a person places their hands on their feet. However, in a T-shape, the hand will be really far from the feet.

**Questions:**

See "weakness".

---

> ### Author Response · Authors · 2025-11-26
> **Response to Reviewer KRMV**
>
> We thank the reviewer for carefully evaluating our paper and providing valuable feedback. We address the concerns in both the common responses and the section below, and we remain open to any further questions or discussion.
>
> **Q1: Eq. (7) encourages close points in the observation space to be close in the canonical space.**
>
> We clarify that Eq. (7) is designed to encourage surface smoothness in the per-frame observation space. In fact, the situation you described is one of the key challenges we aim to address. As illustrated in Figure 1(b), two neighboring pixels in a depth map correspond to two spatially adjacent Gaussians in the observation space. However, as noted in L81–L84, these Gaussians may be far apart in the canonical space due to self-occlusions or holes introduced by motion.
>
> As discussed in L250–L256, we mitigate these issues by filtering out unreliable neighboring Gaussians through the computation of the weight $w_j$. Specifically, when two Gaussians are close in the observation space but far apart in the canonical space, their large canonical space distance results in a small $w_j$. This suppresses the adverse effects of self-occlusion when aggregating depth differences. As also noted by Reviewer Uer2, this smoothness-weight formulation jointly considers geometric relationships in both the canonical and observation spaces, helping preserve meaningful geometric structures.
>
> Additionally, we stop the gradients of $w_j$ and avoid updating canonical Gaussians when computing $\mathcal{L}^r_{smooth}$ to further prevent overfitting. Our ablation study experiments solidly support the effectiveness of our motion-aware weight computation in Eq. (7).

---

### Author Response · Authors · 2025-11-26
**Common Responses to Reviewers**

We thank all reviewers for their detailed feedback and the time committed to evaluating our work. The reviewers recognize the importance of the studied task for a wide range of downstream applications (Reviewer KRMV, Reviewer 3cMT). They find our surface smoothing idea interesting (Reviewer KRMV, Reviewer 3cMT), novel and reasonable (Reviewer 3cMT), and simple yet effective for enforcing geometric regularization (Reviewer Uer2, Reviewer qjx3). The motion-aware modeling is regarded as elegant and interpretable, as it accounts for geometric relationships among neighboring pixels in both canonical and observation spaces (Reviewer Uer2). Reviewers also noted that the paper is well written (Reviewer KRMV), easy to follow (Reviewer 3cMT), and presents promising results (Reviewer KRMV) with improved visual fidelity and more plausible surface reconstructions (Reviewer qjx3), demonstrating broad applicability (Reviewer 3cMT).

Our primary technical contribution lies in integrating surface smoothing into the 3D Gaussian Splatting framework, enabling simultaneous improvements in visual rendering quality and surface synthesis. Specifically, we introduce a novel motion-aware weight computation strategy, a geometry-aware Adaptive Density Control (ADC) mechanism, and the use of virtual-camera supervision to enforce smooth and compact surfaces.

Empirically, we highlight that our method outperforms state-of-the-art approaches in novel view synthesis, novel pose animation, and shape reconstruction within the monocular human modeling setting. In addition to comparisons with six state-of-the-art methods in the original submission, we include two further baselines in this response as suggested. Overall, we conduct extensive evaluations on **twenty-six sequences across five datasets** to ensure statistical significance.

As shown in Tables 2 and 3, different baselines exhibit strengths in different scenarios. For example, LS-Avatar performs second-best across most metrics on MVHumanNet, while 3DGS-Avatar ranks second for novel view synthesis on DNA-Rendering sequences. Across the numerous results presented in the paper and response, our method consistently improves upon existing algorithms in both image rendering and 3D surface reconstruction. In particular, our approach surpasses all baselines in LPIPS with **over 10% relative improvement in nearly all settings** and produces more faithful geometric structures. Ablation studies further demonstrate that motion-aware surface smoothing enhances the reproduction of both realistic shape contours and fine-grained details. We address the remaining reviewer questions individually below.

We hope that the provided clarifications and results, which demonstrate that the claims made in the paper hold over additional baselines, can resolve the remaining concerns. To support reproducibility, we will release the code for our method upon acceptance.

**Comparisons in runtime and model complexity (Reviewer Uer2, Reviewer qjx3).**

Thank you for the constructive comments. As suggested by the reviewers, we follow Table 7 of ToMie (Zhan et al., ICCV 2025) and report the corresponding results below. Specifically, we provide training time and rendering speed to illustrate computational efficiency, as well as the number of learnable parameters to assess model complexity. Based on these measurements, we conclude that our method achieves the best overall performance while maintaining computational costs comparable to state-of-the-art models.

|                            | iHuman | ExAvatar | GauHuman | GoM-Avatar |  Dyco | 3DGS-Avatar | LS-Avatar |  ToMie | Ours |
|:--------------------------:|:------:|:--------:|:--------:|:----------:|:-----:|:-----------:|:---------:|:------:|:----:|
|        Training Time       |  3min  |    3h    |  25 min  |     2d     |  12h  |    30 min   |    15h    | 30 min |  1h  |
|    Rendering Speed (FPS)   |   70+  |    20    |   180+   |     30+    |  0.2  |     80+     |    0.05   |   60+  |  60+ |
| Parameter Number (Million) |  14.33 |   4.14   |    1.1   |    0.95    | 97.49 |     1.91    |    1.52   |  1.22  |  1.7 |

Note that, except for LS-Avatar which requires two V100 GPUs for training, all other methods are trained and evaluated using a single RTX 3090 GPU.

---

> ### Author Response · Authors · 2025-11-26
> **More Results - Part 1**
>
> **Limited improvement from the contributions (Reviewer KRMV, Reviewer 3cMT)**
>
> We emphasize that the majority of the performance gains stem from our proposed surface smoothing mechanism, rather than from replacing 3DGS with GoF or incorporating Sapiens-based normal-map supervision. As summarized in Section 4.3, our core contributions yield significant improvements on top of utilizing the state-of-the-art GoF-based rendering and Sapiens-based normal supervision.
>
> Please note that the original descriptions in Section 4.3 may have caused unnecessary confusion. To clarify, the seven ablated models are constructed sequentially by progressively removing components. Each model is built on top of the preceding one.
> For example, the **$\mathrm{w/o\ \mathcal{L}_{sap}}$** in (6) is obtained by removing $\mathcal{L}_{sap}$
>  from the **$\mathrm{w/o\ \mathcal{L}_{sap}}$** configuration in (5) which have fully disenabled the effects of surface smoothing. We will revise the manuscript to make this ordering explicit.
>
> To better highlight the contribution of each component, we have revised the naming of the ablation variants and report their results in the table below. Specifically, **only 3DGS** refers to training the network solely within the 3D Gaussian Splatting framework, while **with GoF** indicates that the rendering operator is replaced with GoF which equals to the **$\mathrm{w/o\ \mathcal{L}_{sap}}$** in Figure 5. Additionally,  **GoF + Sapiens** denotes the configuration that only incorporates both the GoF renderer and Sapiens’ normal-map supervision which equals to the $\mathrm{w/o\ \mathcal{L}_{smooth}}$ configuration.
>
> The results clearly show that the full model equipped with surface smoothing, consistently outperforms its ablated counterparts, demonstrating the effectiveness of our approach.
>
> |            | PSNR  | SSIM   | LPIPS  |
> |:----------:|:-----:|:------:|:------:|
> | Novel View |       |        |        |
> | only 3DGS   | 24.73 | 0.963  | 40.51  |
> | with GoF   | 24.80 | 0.964  | 39.95  |
> | GoF + Sapiens | 24.98 | 0.964 | 38.57  |
> | Full Model | **25.30** | **0.968** | **32.10**  |
> | Novel Pose |       |        |        |
> | only 3DGS   | 22.95 | 0.954 | 51.19 |
> | with GoF   | 22.96 | 0.954 | 51.17 |
> | GoF + Sapiens | **23.03** | 0.954 | 50.05 |
> | Full Model | 23.00 | **0.957** | **44.56** |
>
> Correspondingly, Figure 5 and Figure B show that using only Sapiens’ normal supervision and the GoF operator is insufficient to produce visually pleasing renderings or geometrically plausible results. In contrast, incorporating surface smoothing leads to more faithful image reconstruction and smoother, more coherent surface geometry with minimal artifacts. Both qualitative and quantitative evaluations reveal the empirical significance of the proposed surface smoothing.

---

> ### Author Response · Authors · 2025-11-26
> **More Results - Part 2**
>
> **Results with more baselines (Reviewer qjx3, Reviewer 3cMT)**
>
> Thank you for the constructive comments. We evaluate our method with two additional baselines including iHuman (ECCV 2024) and Expressive Gaussian Avatar (ECCV 2024) to further validate the importance of the motion-aware surface smoothing. For each baseline, we achieve results through their released source code with public hyper-parameters as specified in their papers.
>
> We illustrate the comparison examples in Figure A of the uploaded revision. As reported in the table below and detailed in Table B and Table D, our method achieves better quantitative results than both baselines in both novel view synthesis and novel pose rendering. According to Table 1 of the ExAvatar paper, we outperform ExAvatar and thus in turn outperform HUGS (Kocabas et al., 2024).
>
>
> |            | PSNR  | SSIM   | LPIPS  |
> |:----------:|:-----:|:------:|:------:|
> | Novel View |       |        |        |
> | iHuman  | 22.09 | 0.954 | 61.74 |
> | ExAvatar | 23.56 | 0.961 | 40.07  |
> | Ours | **25.30** | **0.968** | **32.10** |
> | Novel Pose |       |        |        |
> | iHuman  | 21.34 | 0.949 | 68.66 |
> | ExAvatar | 21.98 | 0.954 | 49.92 |
> | Ours | **23.00** | **0.957** | **44.56** |

---

### Author Response · Authors · 2025-11-26
**Revision Uploaded**

We thank the reviewers and ACs for their time and constructive feedback. In response, we have revised the paper and uploaded a new version with all modifications highlighted in blue. Below, we summarize the key changes made to the main paper and appendix.

1. Adding Figure A to provide visual comparisons for the iHuman and ExAvatar baselines.
2. Adding two tables in Table B and Table D to make the requested quantitative comparisons with iHuman and ExAvatar on eight MVHumanNet sequences.
3. Updating Section 4.3 to describe the formulations of the ablated $w/o\ \{w_j\}$ and "Canonical Smoothing" variants.
4. Updating Table 1 to add the results of the $w/o\ \{w_j\}$ and the "Canonical Smoothing" models to better reveal the effectiveness of motion-aware weight computation.
5. Adding Figure B to provide visual examples for the ablated $w/o\ \{w_j\}$ and "Canonical Smoothing" models.
6. Adding Table F to compare the surface reconstruction quality with baselines in terms of Chamfer Distance and normal consistency, highlighting the improved shape detail.
7. Adding Table G in the appendix to compare with baselines in terms of training time, rendering speed and numbers of learnable parameters.
8. Adding Figure L as failure case examples to provide more contexts of our framework’s limitations.
9. Updating the supplementary video to make the visualization more stable and add the generation examples of the whole human character from 360$\degree{}$ view visualization.
10. Correcting typos and improving wording as suggested by the reviewers.

---

### Comment · Area_Chair_iCE4 · 2025-11-26
**Discussion with Authors**

Dear Reviewers,

The authors have diligently provided responses to your questions and concerns. I request you to please review the authors' responses, acknowledge that you have read them and actively engage with them in further discussion as needed.

This discussion period, with the authors, will end on December 2, 2025 (AoE). However, I request that you not wait until the last minute and actively engage with the authors early.

Best, AC

---

> ### Author Response · Authors · 2025-11-28
> **Thanks for the reviews**
>
> Dear AC and Reviewers,
>
> First let us greatly thank AC for this message. Hope you all have a very nice Thanksgiving day! And we also thank the reviewers for your time and thoughtful evaluation of our paper, as well as for the many constructive suggestions, from clarifying specific details to recommending comparisons with more recent methods. We greatly appreciate your feedback. During the author–reviewer discussion period, we carefully addressed each comment and substantially strengthened our experimental section by adding the comparison results and ablation studies you suggested. With these additional experiments, we can more confidently demonstrate both the empirical superiority of our method over state-of-the-art baselines and the effectiveness of our key contributions. We also commit to releasing the code upon acceptance.
>
> With the updated manuscript, we kindly invite the reviewers to re-evaluate our submission, and we remain open to further discussion.

---

### Author Response · Authors · 2025-12-03
**Comparison of Experimental Settings**

To further validate the comprehensiveness of our evaluation protocol, we provide a detailed comparison respecting the experimental settings below.

|                         |                                                    Chosen datasets for image synthesis                                                   | Sequence Numbers |                                                                                                Chosen baselines                                                                                               | Baseline Number |
|:-----------------------:|:----------------------------------------------------------------------------------------------------------------------------------------:|:----------------:|:-------------------------------------------------------------------------------------------------------------------------------------------------------------------------------------------------------------:|:---------------:|
|   GauHuman (CVPR 2024)  |                            MonoCap (CVPR 2020+ACM ToG 2021), ZJU_MoCap (CVPR 2021), DNA-Rendering (ICCV 2023)                            |        12        | PixelNeRF (CVPR 2021), Neural Body (CVPR 2021), Anim-NeRF (ICCV 2021), NHP (NeurIPS 2021), AniSDF (arXiv 2022), HumanNeRF (CVPR 2022), DVA (SIGGRAPH 2022), InstantNVR (CVPR 2023), InstantAvatar (CVPR 2023) |        9        |
|  GoM-Avatar (CVPR 2024) |                                     PeopleSnapshot (CVPR 2018), ZJU-MoCap (CVPR 2021), YouTube video                                     |        12        |                                  Anim-NeRF (arXiv 2021), NeuralBody (CVPR 2021), HumanNeRF (CVPR 2022), NeuMan (ECCV 2022), MonoHuman (CVPR 2023), InstantAvatar (CVPR 2023)                                  |        6        |
| 3DGS-Avatar (CVPR 2024) |                                             PeopleSnapshot (CVPR 2018), ZJU-MoCap (CVPR 2021)                                            |        10        |                                                NeuralBody (CVPR 2021), HumanNeRF (CVPR 2022), ARAH (ECCV 2022), MonoHuman (CVPR 2023), Instant-NVR (CVPR 2023)                                                |        5        |
|     Dyco (ECCV 2024)    |                                               I3D-Human (ECCV 2024), ZJU-MoCap (CVPR 2021)                                               |        13        |                                              Anim-NeRF (arXiv 2021), NeuralBody (CVPR 2021), AniSDF (arXiv 2022), HumanNeRF (CVPR 2022), 3DGS-Avatar (CVPR 2024)                                              |        5        |
|   ExAvatar (ECCV 2024)  |                                                 NeuMan (ECCV 2022), X-Humans (CVPR 2023)                                                 |         9        |                      HumanNeRF (CVPR 2022), NeuMan (ECCV 2022), InstantAvatar (CVPR 2023), Vid2Avatar (CVPR 2023), GaussianAvatar (CVPR 2024), 3DGS-Avatar (CVPR 2024), HUGS (CVPR 2024)                      |        7        |
|    iHuman (ECCV 2024)   |                          PeopleSnapshot (CVPR 2018), UBC-Fashion (BMVC 2019), Multi-Garment dataset (ICCV 2019)                          |        10        |                                                                      Anim-NeRF (arXiv 2021), InstantAvatar (CVPR 2023), GART (CVPR 2024)                                                                      |        3        |
|  LS-Avatar (ICLR 2025)  | MonoPerfCap (SIGGRAPH 2018), ZJU-Mocap (CVPR 2021), SynWild (CVPR 2023), ActorsHQ (SIGGRAPH 2023), MVHumanNet (CVPR 2024), Youtube video |        24        | HumanNeRF (CVPR 2022), MonoHuman (CVPR 2023), Vid2Avatar (CVPR 2023), PoseVocab (SIGGRAPH 2023), NPC (ICCV 2023), PM-Avatar (ICLR 2024), 3DGS-Avatar (CVPR 2024), GoMAvatar (CVPR 2024), GauHuman (CVPR 2024) |        9        |
|    ToMie (ICCV 2025)    |                                             ZJU-MoCap (CVPR 2021), DNA-Rendering (ICCV 2023)                                             |        14        |                                                           Im4D (SIGGRAPH Asia 2023), 3DGS-Avatar (CVPR 2024), GART (CVPR 2024), GauHuman (CVPR 2024)                                                          |        4        |
|           Ours          |          MonoPerfCap (SIGGRAPH 2018), ActorsHQ (SIGGRAPH 2023), DNA-Rendering (ICCV 2023), MVHumanNet (CVPR 2024), Youtube video         |        26        |                  GauHuman (CVPR 2024), GoM-Avatar (CVPR 2024), 3DGS-Avatar (CVPR 2024), Dyco (ECCV 2024), ExAvatar (ECCV 2024), iHuman (ECCV 2024), LS-Avatar (ICLR 2025), ToMie (ICCV 2025)                  |        8        |

This table shows that we evaluated under the most comprehensive settings among all baselines, including the latest methods and the widest range of recent datasets. This breadth together with strong empirical gains across all datasets reveals that our approach advances SOTAs in both rendering and shape reconstruction.

---

### Author Response · Authors · 2025-12-03
**Complete Evaluation of Images, Surfaces, and Runtime**

Since different reviewers commented on different aspects of our evaluation, from geometric reconstruction to computational cost, we consolidate the relevant results here. Below, we present detailed quantitative comparisons on eight MVHumanNet sequences covering novel view synthesis, novel pose animation, surface reconstruction, and model efficiency, all of which are standard metrics in prior work. These results provide a comprehensive positioning of our method within the current research landscape. The best scores are highlighted in bold.

|                         |  DyCo | LS-Avatar | GauHuman | GoM-Avatar |  ToMie | 3DGS-Avatar | iHuman | ExAvatar |    Ours   |
|:-----------------------:|:-----:|:---------:|:--------:|:----------:|:------:|:-----------:|:------:|:--------:|:---------:|
|   `Novel View Synthesis` |       |           |          |            |        |             |        |          |           |
|           PSNR          | 22.72 |   23.14   |   24.42  |    23.73   |  24.20 |    24.39    |  22.09 |   23.56  | **25.30** |
|           SSIM          | 0.954 |   0.953   |   0.961  |    0.963   |  0.960 |    0.963    |  0.954 |   0.961  | **0.968** |
|          LPIPS          | 51.03 |   40.65   |   47.58  |    42.25   |  51.81 |    41.25    |  61.74 |   40.07  | **32.10** |
|   `Novel Pose Animation`  |       |           |          |            |        |             |        |          |           |
|           PSNR          | 20.98 |   22.24   |   22.79  |    22.26   |  22.53 |    22.81    |  21.34 |   21.98  | **23.00** |
|           SSIM          | 0.947 |   0.954   |   0.954  |    0.953   |  0.952 |    0.954    |  0.949 |   0.954  | **0.957** |
|          LPIPS          | 68.21 |   49.82   |   57.43  |    53.85   |  62.09 |    51.47    |  68.66 |   49.92  | **44.56** |
| `Geometry Reconstruction` |       |           |          |            |        |             |        |          |           |
|  Chamfer Distance (CD)  |  6.02 |    5.18   |   14.17  |    7.30    |  11.71 |    12.00    |  13.35 |   21.35  |  **4.14** |
| Normal Consistency (NC) | 0.707 |   0.786   |   0.663  |    0.768   |  0.678 |    0.652    |  0.568 |   0.603  | **0.803** |
| `Efficiency & Complexity` |       |           |          |            |        |             |        |          |           |
|      Training Time      |  12h  |    15h    |  25 min  |     2d     | 30 min |    30 min   |  **3min**  |    3h    |     1h    |
|  Rendering Speed (FPS)  |  0.2  |    0.05   |   **180+**   |     30+    |   60+  |     80+     |   70+  |    20    |    60+    |
|   Parameter Number (M)  | 97.49 |    1.52   |    1.1   |    **0.95**    |  1.22  |     1.91    |  14.33 |   4.14   |    1.7    |

We observe that our surface-smoothing strategy consistently outperforms all baselines across novel view synthesis, novel pose animation, and surface reconstruction, while maintaining reasonable training efficiency, competitive rendering speed, and comparable model complexity. These results validate the empirical effectiveness and broad applicability of our approach. As suggested by Reviewer qjx3, we also report mesh reconstruction scores for iHuman and ExAvatar, two recent mesh-based human reconstruction methods, to make our experimental evaluation more complete.

---

### Author Response · Authors · 2025-12-04
**Thank you for the thorough reviews**

In the end of the author discussion phase, we would like to begin by sincerely thanking the Reviewers and the ACs for the valuable feedback that has helped us significantly improve our paper. In response to your insightful comments, we carefully addressed every question and substantially strengthened our submission by providing more comprehensive evaluation results and offering additional clarifications, as summarized earlier.

We regret that we did not have the opportunity for deeper, more detailed discussions with the reviewers this year. Nevertheless, we would like to reiterate the core strengths of our method.
(1) Conceptually, our framework is simple, intuitive, novel and effective. Our main contributions are the introduction of a motion-aware regularization term, a geometry-aware Adaptive Density Control (ADC) module, and the utilization of virtual cameras.
(2) Experimentally, we conduct an extensive evaluation, comparing against eight of the latest and most representative baselines across 26 sequences spanning diverse scenarios.

With the newly added experiments, together with the results presented in the original submission, we are now more confident in our scientific conclusion: `Surface smoothing provides consistent and significant improvements over existing baselines in novel view synthesis, novel pose animation and shape modeling across a wide range of empirical settings.`

In line with observations reported in prior work (e.g., Sec. 5.2 of 3DGS-Avatar and Sec. 4.2 of LS-Avatar), we find that under monocular settings, perceptual metrics (e.g., LPIPS) are more informative than pixel-wise ones such as PSNR, which can be sensitive to slight output misalignments and varying outdoor lighting. Across all such perceptual metrics including LPIPS, KID, and FID, we consistently outperform the second-best methods with more than 10\% relative improvement, a level of gain that prior papers also regard as substantial.

Finally, we would like to express our sincere gratitude to the ACs for their service to the research community, especially in a year that has been unpredictable and challenging. Without your dedication, professionalism, and commitment to maintaining the quality of the academic ecosystem, both the efforts of many researchers and the reputation of the conference would be at risk. We hope to continue learning from the valuable insights provided by reviewers and ACs in the future.

---

### Meta-Review · Area_Chair_hYx7 · 2026-01-02

**Summary:**

Most reviewers initially leaned toward rejecting this paper in the first round (2644), but the authors effectively addressed most concerns. The ACs identified the main issue: "who plays the leading role, sapiens vs. smoothing." Since the motion-aware smoothing term is a key contribution, this debate is crucial. The authors explained that "the seven ablated models are constructed sequentially by progressively removing components," which aligns with the clear monotonic trend in Table 1.

However, despite this clarification, the paper lacks novelty; only $L_{smooth}$ stands out as a solid contribution, while other terms come from existing techniques. Although not as critical as $L_{smooth}$, Sapiens remains essential according to Figure 5. The poorer performance in ActorHQ (novel pose) highlights limitations of the smoothing term and raises concerns about generalization with non-rigid clothing deformations.

Given these points, the AC believes this paper falls below ICLR's standards and recommends rejection. The authors should incorporate all feedback to improve the manuscript for future submissions.

**Reviewer Concerns:**

- Reviewer KRMV: The concern about Eq(7) is well addressed. The AC agrees that this smooth design refines regions close in both posed and canonical space, and the "sapiens vs. smoothing" issue is effectively tackled in "More results - Part 1."

- Reviewer Uer2: All concerns are well addressed, supporting the effectiveness of the motion-aware smoothing term.

- Reviewer qjx3: Most concerns are addressed, but the issue with ActorHQ (novel pose) highlights a limitation of the smoothing regularization, which may affect complex geometric structures like non-rigid clothing dynamics. Both ToMiE and GauHuman do not incorporate physical simulation of garments yet achieve better results in ActorHQ (novel pose).

- Reviewer 3cMT: All concerns are well addressed.

**Reviewer Scores:**

- Reviewer KRMV (conf 4): Likely to raise the rating (4).

- Reviewer Uer2 (conf 3): Likely to maintain the rating (6).

- Reviewer qjx3 (conf 4): Likely to keep the rating (4).

- Reviewer 3cMT (conf 4): Likely to raise the rating (6).

---

### Decision · Program_Chairs · 2026-01-26

Reject